# ADMIRE-BayesOpt: Accelerated Data MIxture RE-weighting for Language Models with Bayesian Optimization

**Shengzhuang Chen**[*]  *shengzhuang.chen@thomsonreuters.com*
*Thomson Reuters Foundational Research &*
*Imperial College London*

**Xu Ouyang**[*]  *ftp8nr@virginia.edu*
*University of Virginia*

**Michael Arthur Leopold Pearce**  *michaelp@graphcore.ai*
*Graphcore*

**Thomas Hartvigsen**  *hartvigsen@virginia.edu*
*University of Virginia &*
*Thomson Reuters Foundational Research*

**Jonathan Richard Schwarz**  *jonathan.schwarz@thomsonreuters.com*
*Thomson Reuters Foundational Research &*
*Imperial College London*
[*]*: Equal contribution.*

**Reviewed on OpenReview:** *https://openreview.net/forum?id=0Euvm9zDpu*

## Abstract

Determining the optimal data mixture for large language model training remains a challenging problem with an outsized impact on performance. In practice, language model developers continue to rely on heuristic exploration since no learning-based approach has emerged as a reliable solution. In this work, we propose to view the selection of training data mixtures as a black-box hyperparameter optimization problem, for which Bayesian Optimization is a well-established class of appropriate algorithms. Firstly, we cast data mixture learning as a sequential decision-making problem, in which we aim to find a suitable trade-off between the computational cost of training exploratory (proxy-) models and final mixture performance. Secondly, we systematically explore the properties of transferring mixtures learned at a small scale to larger-scale experiments, providing insights and highlighting opportunities for research at a modest scale. By proposing Multi-fidelity Bayesian Optimization as a suitable method in this common scenario, we introduce a natural framework to balance experiment cost with model fit, avoiding the risks of overfitting to smaller scales while minimizing the number of experiments at high cost. We present results for pre-training and instruction finetuning across models ranging from 1 million to 7 billion parameters, varying from simple architectures to state-of-the-art models and benchmarks spanning dozens of datasets. We demonstrate consistently strong results relative to a wide range of baselines, resulting in **speed-ups of over 500%** in determining the best data mixture on our largest experiments. In addition, we broaden access to research by sharing *ADMIRE IFT Runs*, a dataset of 460 full training & evaluation runs worth over **13,000 GPU hours**, greatly reducing the cost of conducting research in this area. Finally, we highlight rich opportunities for future research in this area, helping bridge the gap towards a comprehensive understanding of the broader effects of training data on model generalization.

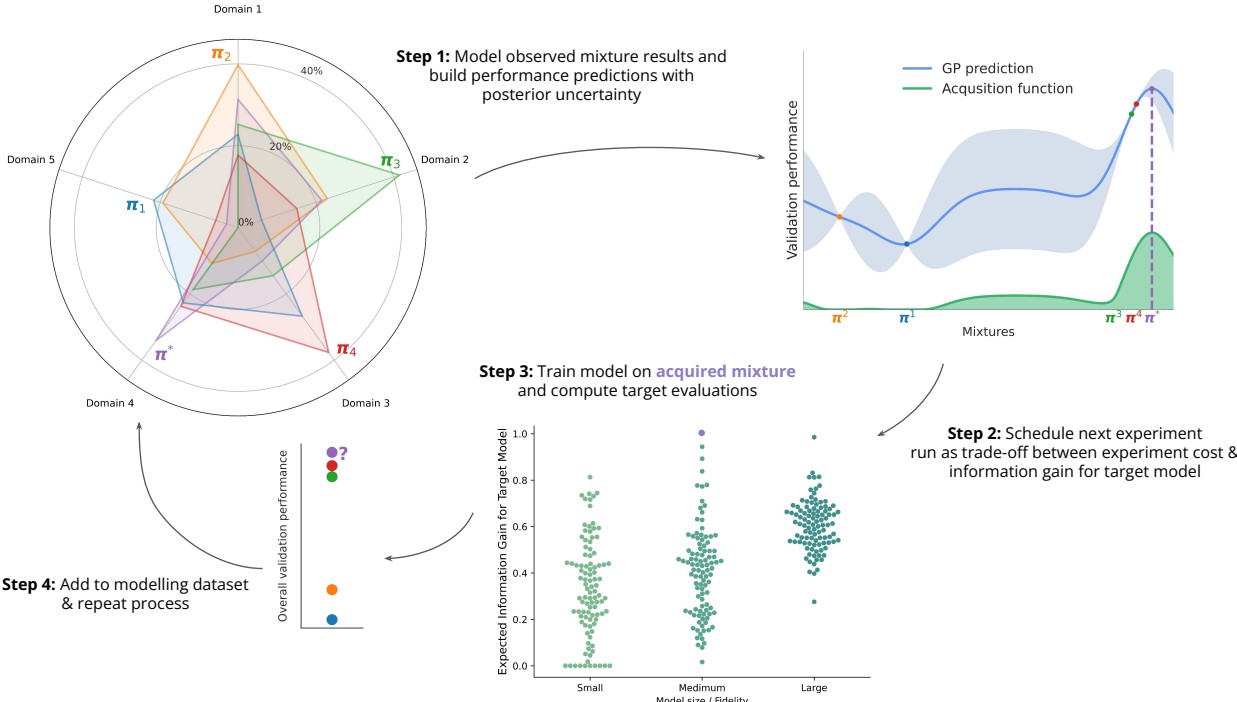

Figure 1: An Overview of our method. We model the contribution of training domains to a target evaluation with a Gaussian Process. By finding the maximum of an acquisition function that provides a numeric trade-off between exploration and exploitation, ADMIRE-BayesOpt rapidly finds mixtures that outperform common practices such as random exploration.

# 1 Introduction

Much of the scientific literature in Deep Learning over the last few decades was firmly anchored in the discovery methods focused on improving learning through algorithms and architectures. Recent years, however, have seen a convergence around a small subset of well-established techniques (Rumelhart et al., 1985; Kingma & Ba, 2014; Vaswani et al., 2017), which have shown remarkable resilience despite many attempts to challenge their status. As a result, the field has witnessed a shift towards ideas focused on orthogonal improvement, among which a new focus on data-centric ideas has emerged as a vibrant research field. This shift has been particularly pronounced in the development of Large Language Models (LLMs) (e.g Devlin et al., 2019; Brown et al., 2020). The importance of understanding how data impacts the quality of a trained language model during both pre- and post-training is evidenced by a rich body of literature (see (Albalak et al., 2022) for an overview), a plethora of comments in published reports, as well as academic workshops (e.g. dml, 2024) and data competitions centred around data selection for language models (e.g. Li et al., 2024). Tech reports from major industrial investors in LLMs openly state *"We find that data quality is an important factor for highly-performing models, and believe that many interesting questions remain around finding the optimal dataset distribution for pre-training."* (Google, 2023) or *"In each round of post-training, we adjust our overall data mix carefully across these axes to tune performance across a wide range of benchmarks. Our final data mix epochs multiple times on some high-quality sources and downsamples others."* (Grattafiori et al., 2024). Finally, in the open-source post-training project Tülu 3 (Lambert et al., 2024), the authors show how optimising the mixture of data can improve the average performance across a number of challenging benchmarks by over 10%, an improvement that would otherwise constitute a major algorithmic breakthrough.

---

Dataset: *ADMIRE IFT Runs* ⬡ ADMIRE-BayesOpt Code

Since datasets for LLM training are typically assembled from various domains (e.g. The Pile (Gao et al., 2020) being a mixture of web data, Wikipedia, Github, News etc.), the final data composition can be seen as a mixture of different sources, each weighted by a factor corresponding to its contribution to the final mix. While optimizing these weights heuristically (Chowdhery et al., 2023; Touvron et al., 2023; Lambert et al., 2024) remains common practice, exhaustively searching data mixtures is prohibitively expensive. On the other end of the spectrum, methods that directly learn the data mixture with a smaller proxy model (e.g. Xie et al., 2023) and perform zero-shot transfer to larger scale run risk of over-fitting the mixture to the more modest proxy-model capabilities, thereby overweighting simple to medium difficulty examples while undersampling complex reasoning cases that escape the capabilities of a small model.

We propose **ADMIRE-BayesOpt**: **A**ccelerated **D**ata **MI**xture **RE**-weighting with **Bayes**ian **Opt**imization (Figure 1). Our key insight is that the extensive framework for sequential decision making and Bayesian Optimization provides a natural paradigm for data-mixture learning. First, following prior work (Liu et al., 2024), ADMIRE-BayesOpt casts re-weighting as a regression problem from domain weights to a target evaluation metric. In this regression formulation, a single data point $(\boldsymbol{\pi}, y)$ corresponds to the weights of the mixture, i.e. a point on the simplex $\sum_i \pi_i = 1$ where $\pi_i \in [0, 1]$ gives the weight of a source domain, and target evaluation metric $y$. A dataset of experimental results $\mathcal{D} = \{(\boldsymbol{\pi}_t, y_t)\}$ can thus directly allow the prediction of performance for a new data mixture. Since each element in $\mathcal{D}$ corresponds to a full LLM training run, making $\mathcal{D}$ very expensive to obtain, we carefully balance exploration and exploitation in this space by explicitly modelling the regression uncertainty through Gaussian Processes (Williams & Rasmussen, 2006). Second, by finding a trade-off between mean predicted performance and uncertainty using acquisition functions (Jones et al., 1998; Wang & Jegelka, 2017a), we explore the space in a principled and efficient manner, leading to significantly accelerated convergence and improved results when compared to a range of existing methods.

The benefits of introducing black-box sequential decision making are not limited to faster convergence in an otherwise established paradigm. It is common practice to use cheaper proxy models to experiment with various mixtures (i.e, collect $\mathcal{D}_{proxy}$) and then apply the same mixture at a larger scale. Besides, recent data mixture scaling law papers (Liu et al., 2024; Ye et al., 2024) propose empirical function formulas of $\mathcal{D} = \{(\boldsymbol{\pi}_t, y_t)\}$. We thus introduce Multi-Fidelity Bayesian Optimisation Kandasamy et al. (2017); Forrester et al. (2007); Takeno et al. (2020), which, contrary to prior work, allows a principled trade-off between collecting results at different model sizes and the expected ability to correctly predict mixtures that generalise across scale. This reduces the reliance on a proxy model or a simple law formula, whereas zero-shot transfer relies on a strong proxy model.

We demonstrate these properties of **ADMIRE-BayesOpt** through experiments on pre-training and instruction-finetuning (IFT) across a variety of model sizes. We start with the more modest pre-training experiments introduced by (Liu et al., 2024) on The Pile (Gao et al., 2020) and then scale to modern representative post-training workloads on the Tülu 3 dataset collection (Lambert et al., 2024) using the Qwen 2.5 (Yang et al., 2024b) family of pre-trained models. To facilitate further research, we further contribute the ADMIRE-BayesOpt collection, which includes all training artifacts for over 460 IFT runs for 0.5b, 3b, and 7b Qwen 2.5 models, each being trained on 200k examples from the Tülu 3 dataset. This extensive collection provides significant opportunities for researchers, especially when limited by computational constraints.

While we present full experimental results in Section 6, Figure 2 provides a preview of our results in both the standard Bayesian Optimization (BO) and Multi-Fidelity BO settings. Figure 2a shows the performance of a 7b model trained on a recommended mixture discovered using larger-scale (7b) or smaller-scale experiments (500m) only. In comparison to the recent work by (Liu et al., 2024), we show impressive speed-ups ( 195% when comparing the time to find the optimal mixture, 1500% when comparing the time needed for (Liu et al., 2024) to match the best performance achieved by ADMIRE-BayesOpt when running only small models). Figure 2b shows the extension to multi-fidelity settings (automating the choice of what experiments to run), highlighting that our method learns to schedule in three phases of training runs, achieving close to state-of-the-art performance after running almost exclusively small-scale experiments. Compared to the speed-up figures previously mentioned, the gap to (Liu et al., 2024) increases to a speed-up of over 500% measured as the time needed to find the best mixture for larger models.

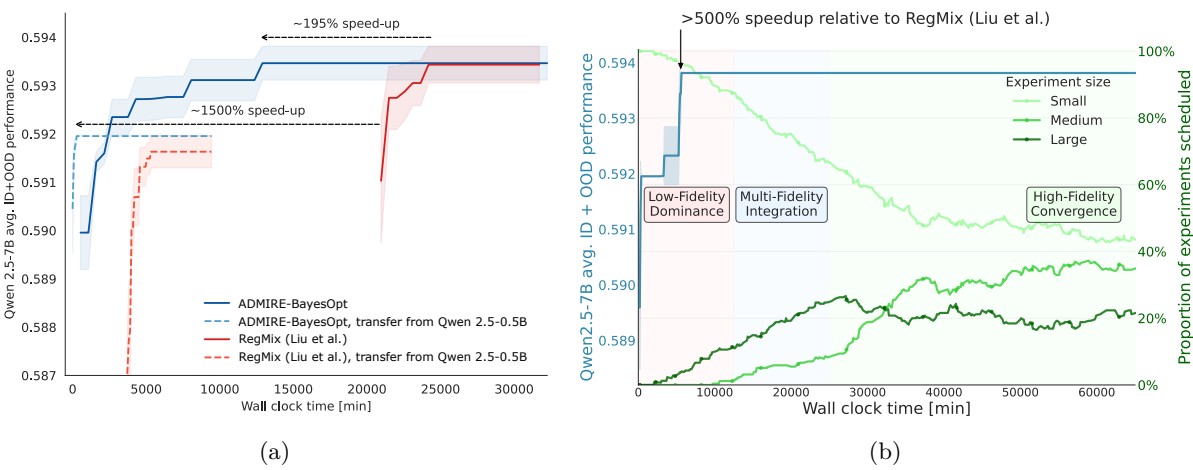

Figure 2: Results obtained running our data mixture optimization pipeline. (a) ADMIRE-BO in comparison to (Liu et al., 2024) on the Tülu 3 SFT dataset. Shown is the performance of a Qwen-2.5 7B model when trained on a discovered mixture. (b) Experiment scheduling and performance for ADMIRE-MFBO. Experimental runs are divided into broadly three phases.

## 2 Related Work

**Data Mixture Optimization** A significant number of recent works have noticed the potential for careful data mixture optimization to both accelerate training and improve performance. The work by Sorscher et al. (2022) may be credited for further popularizing research in this direction by making a key argument of how data-centric techniques can beat well-studied scaling laws (Kaplan et al., 2020; Hoffmann et al., 2022). We can broadly categorize these data optimization methods as data preprocessing and training-aware methods. Data preprocessing techniques aim to identify the best subset ahead of training, while training-aware methods typically aim to find a suitable batch of data for each learning step. Nevertheless, general ideas of how difficulty could be identified are typically shared, and several key ideas can be found in approaches that focus on either direction. While not explicitly learning the mixture coefficient of a source, these techniques result in an implicit mixture as data points are typically prioritized non-uniformly. While we focus on applications to LLMs, several of the approaches below focus on vision applications, although their methods are general enough to extend to most learning domains. For an LLM-specific perspective, see (Marion et al., 2023).

Direct learning statistics such as the training loss, gradient, or the perplexity (in case of LLMs) (Paul et al., 2021; Tack et al., 2023; Ankner et al., 2024) are readily available during training and provide a conceptually simple signal. Other techniques focus on the well-studied phenomenon of memorization (e.g. Carlini et al., 2022; Biderman et al., 2023) in deep learning, with a particular approach (Feldman & Zhang, 2020) defining memorization as the difference in probability of predicted the correct label for an example depending on whether or not this example is in the training data. Intuitively, a low memorization score suggests easy example redundancy with the rest of the data, albeit this idea is sensitive to noisy examples. A downside of such metrics is that they typically compute a difficulty score but leave the choice of whether to train on simple examples, difficult ones, or a mixture thereof as an additional choice with a large impact on performance.

More explicit data optimization methods focus on semantic de-duplication, typically by comparing examples in the feature space of a reference model, (e.g. Abbas et al., 2023) or computing the alignment of low-rank training example gradients with those on a held-out set (Xia et al., 2024), allowing direct targeting towards a specific generalization evaluation. While this is appealing in principle, the inherent limitation on LLM evaluations may result in the risk of removing generally useful examples that are, however, not directly measured by the skills represented by the held-out validation set.

A key idea initially introduced in (Mindermann et al., 2022) and later built upon in various follow-up works (e.g. Evans et al., 2024; Brandfonbrener et al., 2024) is the concept of *learnability scores* wrt. a reference

model. This set of metrics defines the importance score of an example as the difference in loss of a currently training model and a trained and frozen reference model. Intuitively, this has the benefit of an online measure of examples that are learnable but have not yet been learned, and provides some robustness to noisy examples while avoiding the need to define what level of difficulty to prioritize.

**Data-Mixture Re-Weighting methods**   There are two primary approaches to identifying the optimal mixture of data domains: proxy model-based methods and law-based methods. DoReMi (Xie et al., 2023), a representative proxy-based approach, trains two small models to optimize domain weights for a large target model. First, a reference model is trained on unoptimized domain weights to simulate the behavior of the target model. Then, a proxy model is trained based on this reference model to optimize the domain reweighting. Finally, the large target model is trained on the reweighted dataset. However, such methods rely heavily on the assumption that small proxy models accurately reflect the data preferences of the larger target model. In contrast, our BO-based approach is more efficient and flexible, as it does not require proxy model training.

More recently, law-based approaches such as RegMix (Liu et al., 2024) and Data Mixing Law (Ye et al., 2024) have been proposed, claiming to outperform DoReMi. These methods model the relationship between data mixtures and performance metrics using explicit mathematical functions. RegMix assumes a linear relationship, while Data Mixing Law introduces an exponential function following linear interactions. However, modeling such a complex relationship using a fixed, empirical functional form is inherently limited. In contrast, our BO framework avoids explicit assumptions by treating the mixture-metric relationship as a black-box function, enabling it to model complex interactions more effectively and flexibly.

**Bayesian Optimization**   The relationship between dataset mixture weights $\boldsymbol{\pi}$ and the validation score $y$ on a downstream benchmark/task involves training an LLM training followed by inference on a validation set and measuring performance. Hence the relationship between $\boldsymbol{\pi}$ and $y$ does not have a practical analytical form nor gradient information and so one may treat it as a black box. However we may still assume smoothness, small changes in $\boldsymbol{\pi}$ yield small changes in $y$, and hence we may build a prediction model. Such problems arise in many domains, simulation optimization (Pasupathy & Henderson, 2011; Eckman et al., 2023), physics and nuclear reactor design (Ginsbourger et al., 2014; Char et al., 2019), robotics control (Martinez-Cantin et al., 2009; Lechuz-Sierra et al., 2024). These problems have relatively low dimensional input, up to 20, further one may typically assume similar inputs will have similar outputs implying smoothness, and as each data point is expensive, the number of points we can collect is severely limited, up to 1,000. Due to this cost, sequential data collection is more efficient than collecting all points in a single batch as one may incorporate the cumulative knowledge of the points collected so far to determine the next input to evaluate.

Bayesian optimization (BO) has become a go-to method for sequential black box optimization problems. Efficient Global Optimization (Jones et al., 1998) is the standard BO algorithm, the authors proposed to fit a Gaussian process regression model to the black box data then choose the next input to evaluate by optimizing an acquisition function that quantifies the expected benefit (Shahriari et al., 2015; Frazier, 2018).

With the growing cost and complexity of machine learning models and the sensitivity of such models to hyper parameter settings such as learning rate and batch size, early works applied BO to find the best SGD hyper parameters for neural network training (Snoek et al., 2012; Gardner et al., 2014) where each data point requires training a neural network with the given hyper parameters. As a result, BO removes the need for hand crafting or expert tuning.

BO methods have been generalised to problems that require transfer of knowledge from one optimization task to another. For example one has a set of datasets, naively one may use standard BO to independently find optimal parameters for each task, alternatively one may share knowledge across tasks improving data efficiency (Thornton et al., 2013; Bardenet et al., 2013; Poloczek et al., 2016; Pearce et al., 2020).

In this work we consider transferring knowledge from cheap task, small LLM training, to an expensive task, large LLM training. Multi-Fidelity BO (Forrester et al., 2007) methods extends the traditional optimization of an expensive black-box function (high fidelity target) by including a cheaper approximate black box function (low fidelity proxy). Multi-Task BO (Swersky et al., 2013) and Multi-Information Source Optimization

(Poloczek et al., 2017) optimize SGD hyper parameters of an MNIST image classifier and allow the BO algorithm to choose to cheaply train an ML model on a mini dataset (proxy) or the full dataset (target) and show that this outperforms optimizing for the full dataset only. The FABOLAS algorithm (Klein et al., 2017) further treats dataset size as a pseudo-continuous fidelity, the BO algorithm may directly choose how long to train a model for. Follow up work incorporated the popular HyperBand algorithm (Li et al., 2018; Falkner et al., 2017). More recently, the trace-aware Knowledge Gradient (Wu et al., 2020) also treats training iterations as fidelity levels and proposes the Downsampling kernel, that allows the GP to model convergence curves as a monotonic polynomial.

Many language models are released as families of models that vary by parameter count. In this work, we propose to treat LLM parameter count as fidelity level, and data mixture as the variable to optimize, we desire the best full size model while exploiting the ability to train smaller models. By leveraging both sequential data collection and multi-fidelity optimization, we may automate and significantly reduce the compute cost of finding the optimal data mixture for LLM training.

Particular credit should be given to MFMS-GP (Yen et al., 2025), a concurrent work that also applies Multi-Fidelity BO to data mixture optimization. The authors run MFBO over both the training step and model size dimensions, demonstrating the potential of optimizing along these two axes simultaneously. However, their experiments are limited to 1B parameter models with relatively modest training and evaluation setups. In contrast, our work studies Bayesian Optimization for data mixture optimization in a much broader and more realistic setting, including: (1) substantially larger model scales up to 7B parameters; (2) realistic training and evaluation protocols following state-of-the-art practices: our work compares mixture optimization methods based on actual downstream task performance from fully trained models across diverse benchmarks, reflecting realistic LLM development workflows. In contrast, MFMS-GP estimates and compares performance using trained performance predictors rather than actual model evaluation, which may introduce prediction bias and not fully capture the complex relationship between data mixtures and downstream capabilities; and (3) comprehensive coverage of both pre-training and post-training regimes. We include experimental comparisons with MFMS-GP in Section 6, where ADMIRE-BayesOpt consistently demonstrates superior performance, highlighting the efficiency and effectiveness of our approach in practical large-scale scenarios.

## 3 Problem Definition: Data Mixture Optimization

Consider training a target LLM $\tilde{m}$ on a collection of $d$ source datasets. We define a training data mixing ratio as a point on the probability simplex $\boldsymbol{\pi} \in [0,1]^d$ where $\sum_{i=1}^{d} \pi_i = 1$. Given such a mixing ratio and model pair $(\boldsymbol{\pi}, m)$, we train the model using a finite-sized dataset constructed by mixing the $d$ source datasets according to $\boldsymbol{\pi}$, and evaluate the resulting model on a separate set of validation datasets. This yields a scalar validation error $y$ defined as the unweighted average validation error across all validation datasets, i.e., $y = f(\boldsymbol{\pi}, m)$, where $f : \Pi \times \mathcal{M} \to \mathbb{R}$ represents the unknown ground-truth function mapping the complete training-evaluation pipeline to validation performance. The data mixture optimization problem seeks the optimal mixing ratio $\boldsymbol{\pi}^* = \arg\min_{\boldsymbol{\pi} \in \Pi} f(\boldsymbol{\pi}, \tilde{m})$ that minimizes validation error for the target model $\tilde{m}$.

As discussed in Section 1, finding the optimal data mixture is highly non-trivial due to the enormous search space. Existing solutions are insufficient, because they typically rely on trial-and-error approaches, training on multiple handpicked mixtures for evaluation to identify promising candidates or optimize future mixture compositions. This quickly becomes impractical as target model sizes scale. To accelerate practical mixture selection, we formulate the optimization problem through the lens of Bayesian optimization. Here, we aim to sequentially build a parametric Bayesian posterior $\hat{f}(\boldsymbol{\pi}, \tilde{m}) : \Pi \times \mathcal{M} \to \mathbb{R}$ regarding the optimal training data mixture specific to the target model $\tilde{m}$ through rounds of sequential training and evaluation trials gathered from a collection of $M$ (smaller) proxy models, i.e., $\mathcal{M} = \{m_1, \ldots, m_M\}$, that are potentially more computationally efficient for training and evaluation compared to $\tilde{m}$. When $M \geq 2$, this framework is a **Multi-Fidelity Bayesian Optimization** (MFBO) setup, where the acquisition phase can adaptively query and build a Bayesian posterior based on observations across different proxy models. These proxy models serve as information sources with varying computational costs and approximation accuracies, capturing transferability across different model scales and thereby improving sample efficiency. When $M = 1$, this

reduces to a **zero-shot transfer** setting, where we optimize the data mixture using a single proxy model and evaluate the performance of the recommended data mixture on the target model.

Without loss of generality, we assume that training costs vary by proxy model and are known a priori as $\{c_{m_1}, \ldots, c_{m_M}\}$. Given a fixed compute budget $C$ (e.g., dollar cost, GPU hours, FLOPs) accommodating up to $T$ iterations for data mixture optimization, we repeat the following procedure at each step $t \in \{1, 2, ..., T\}$, as illustrated in Figure 1: (**1**) construct a prediction model $\hat{f}_t(\boldsymbol{\pi}, m)$ by fitting a parameterized Bayesian posterior distribution that directly approximates validation performance for any data mixture-model pair based on accumulated observations, $\mathcal{D}_{1:t} = \{(\boldsymbol{\pi}_{t'}, m_{t'}, y_{t'})\}_{t'=1}^{t}$; (**2**) select the next query point $(\boldsymbol{\pi}_{t+1}, m_{t+1})$ by maximizing an acquisition function $\alpha_t^{\texttt{acq}}(\boldsymbol{\pi}_{t+1}, m_{t+1} \mid \hat{f}_t) : \Pi \times \mathcal{M} \to \mathbb{R}$ that balances exploration and exploitation based on the current posterior $\hat{f}_t$; (**3**) train the selected proxy model $m_{t+1}$ with mixture $\boldsymbol{\pi}_{t+1}$, and observe validation errors $y_{t+1} = f(\boldsymbol{\pi}_{t+1}, m_{t+1})$, which consumes $c_{m_{t+1}}$ units of the compute budget; and (**4**) update the predictor with newly acquired observation through Bayesian posterior inference: $\hat{f}_t, (\boldsymbol{\pi}_t, m_{t+1}, y_{t+1}) \to \hat{f}_{t+1}$. As a result, at every step, we can recommend the optimal data mixture for the target model $\tilde{m}$ by exploiting the learned posterior: $\boldsymbol{\pi}^* = \arg\min_{\boldsymbol{\pi} \in \Pi} \hat{f}_t(\boldsymbol{\pi}|\tilde{m})$. Our objective is to identify the data mixture that minimizes validation error on the target model $\tilde{m}$ while minimizing computational cost subject to the budget constraint $C$.

## 4 ADMIRE-BayesOpt

While many approaches could be applied to our formulation of the data mixing problem, we propose Bayesian Optimization and Multi-Fidelity Bayesian Optimization as principled solutions and evaluate their performance against established baselines.

### 4.1 Modelling of Mixture Quality with Gaussian Processes

Given a dataset of mixtures, models, and experiment observations $\mathcal{D}_{1:t} = \{(\boldsymbol{\pi}_{t'}, m_{t'}, y_{t'})\}_{t'=1}^{t}$, we construct a regression model to directly predict validation scores $\hat{y} = \hat{f}(\boldsymbol{\pi}, m)$. Gaussian Process regression is particularly well-suited for this task due to its ability to quantify uncertainty and leverage prior knowledge through its probabilistic framework. A Gaussian Process is characterized by (1) a mean function $\mu_0(\boldsymbol{\pi}, m)$ that encodes prior expectations about validation performance for any given input. This can incorporate domain knowledge—for instance, if historical evidence suggests certain mixtures consistently perform well; and (2) a covariance function $\mathbf{K}((\boldsymbol{\pi}, m), (\boldsymbol{\pi}', m'))$ which governs more abstract properties of the regression model such as smoothness, periodicity, monotonicity over the continuous mixture space $\boldsymbol{\pi}$ and correlations in performance across different language models $m$ and $m'$.

For our MFBO approach, we parameterize the covariance function as a product kernel[1] that separates dependencies over mixtures $\Pi$ and models $\mathcal{M}$

$$\mathbf{K}^{\texttt{MFBO}}((\boldsymbol{\pi}, m), (\boldsymbol{\pi}', m')) = \lambda \mathbf{K}^{\texttt{RBF}}(\boldsymbol{\pi}, \boldsymbol{\pi}') \mathbf{K}^{\texttt{DS}}(m, m'), \tag{1}$$

where $\lambda$ is a scaling parameter, $\mathbf{K}^{\texttt{RBF}}$ is the Radial Basis Function (RBF) kernel, and $\mathbf{K}^{\texttt{DS}}$ is a Downsampling kernel constructed from model parameter counts.

The RBF kernel models the intuition that similar data mixtures should yield similar validation performance. It computes similarity based on Euclidean distance in the mixture space, with correlation decreasing smoothly as mixtures become more dissimilar:

$$\mathbf{K}^{\texttt{RBF}}(\boldsymbol{\pi}, \boldsymbol{\pi}') = \exp\left(-\frac{\|\boldsymbol{\pi} - \boldsymbol{\pi}'\|^2}{2\sigma^2}\right), \tag{2}$$

where $\sigma$ is a learnable length scale parameter. Since mixture proportions lie on the probability simplex, we use a shared length scale across all dimensions to reduce model complexity and mitigate overfitting.

The model kernel $\mathbf{K}(m, m')$ could be specified as another learned $M \times M$ positive-semidefinite matrix, offering maximum flexibility at the cost of $O(M^2)$ hyperparameters. Instead, we leverage the continuous nature

---

[1]https://botorch.org/docs/tutorials/discrete_multi_fidelity_bo/

of model scale by using parameter count as a feature. Specifically, we use the number of language model parameters as the feature rescaled to the range $[0, 1]$ and apply the Downsampling kernel

$$\mathbf{K}^{\mathtt{DS}}(m, m') = c + (1 - s_m)^{1+\delta}(1 - s_{m'})^{1+\delta} \tag{3}$$

where $s_m \in [0, 1]$ represents the normalized parameter count for model $m$. This kernel fits a concave monotonic function of the form $a + b(1 - s_m)^{1+\delta}$ where $a$ and $b$ are inferred by the GP, which encodes the expectation that larger models generally achieve better validation performance—a relationship not captured by the mixture-only RBF kernel. Originally proposed for modeling performance across training iterations (Wu et al., 2020), we adapt it here for model parameter scaling. This parameterization requires learning only two hyperparameters: $c$ and $\delta$.

After conducting $t$ training experiments and observed their outcomes, we have a series of inputs $\tilde{\boldsymbol{\pi}}_{1:t} := \{(\boldsymbol{\pi}_{t'}, m_{t'})\}_{t'=1}^{t'=t}$ and corresponding outputs $\mathbf{y}_{1:t} := \{y_{t'}\}_{t'=1}^{t'=t}$. The posterior predictive distribution $\hat{f}_t(\tilde{\boldsymbol{\pi}})$ for any new mixture-model pair $\tilde{\boldsymbol{\pi}} := (\boldsymbol{\pi}, m)$ follows from standard Gaussian conditioning:

$$\mu(\tilde{\boldsymbol{\pi}}) = \mu_0(\tilde{\boldsymbol{\pi}}) + \mathbf{K}^{\mathtt{MFBO}}(\tilde{\boldsymbol{\pi}}, \tilde{\boldsymbol{\pi}}_{1:t})^\top (\mathbf{K}^{\mathtt{MFBO}}(\tilde{\boldsymbol{\pi}}_{1:t}, \tilde{\boldsymbol{\pi}}_{1:t}) + \sigma_\epsilon^2 \mathbf{I})^{-1} \mathbf{y}_{1:t} \tag{4}$$

$$\mathrm{var}(\tilde{\boldsymbol{\pi}}) = \mathbf{K}^{\mathtt{MFBO}}(\tilde{\boldsymbol{\pi}}, \tilde{\boldsymbol{\pi}}) - \mathbf{K}^{\mathtt{MFBO}}(\tilde{\boldsymbol{\pi}}, \tilde{\boldsymbol{\pi}}_{1:t})^\top (\mathbf{K}^{\mathtt{MFBO}}(\tilde{\boldsymbol{\pi}}_{1:t}, \tilde{\boldsymbol{\pi}}_{1:t}) + \sigma_\epsilon^2 \mathbf{I})^{-1} \mathbf{K}^{\mathtt{MFBO}}(\tilde{\boldsymbol{\pi}}_{1:t}, \tilde{\boldsymbol{\pi}}) \tag{5}$$

where $\mathbf{K}^{\mathtt{MFBO}}(\tilde{\boldsymbol{\pi}}_{1:t}, \tilde{\boldsymbol{\pi}}_{1:t}) \in \mathbb{R}^{t \times t}$ is the Gram matrix over evaluated mixture-model configurations, $\mathbf{K}^{\mathtt{MFBO}}(\tilde{\boldsymbol{\pi}}, \tilde{\boldsymbol{\pi}}_{1:t}) \in \mathbb{R}^{1 \times t}$ contains cross-covariances with observed data, and $\sigma_\epsilon$ is a hyperparameter representing observation noise. To this end, the complete set of hyperparameters $\{\lambda, \sigma, c, \delta, \sigma_\epsilon\}$ is optimized by maximizing the marginal likelihood via gradient-scent (Williams & Rasmussen, 2006).

In the single-fidelity case where $M = 1$, the model kernel $\mathbf{K}^{\mathtt{DS}}(m, m')$ becomes constant and is absorbed into the hyperapameter $\lambda$. Consequently, $\mathbf{K}^{\mathtt{MFBO}}$ in Equation 1 reduces to a standard RBF kernel over the mixture space $\boldsymbol{\pi}$, recovering the standard Bayesian Optimization.

## 4.2 Scheduling New Experiments

Given the surrogate model $\hat{f}_t$ parameterized by the Bayesian posterior in Equation 4 and Equation 5, an acquisition function quantifies the expected utility of evaluating a candidate mixture and measuring its validation performance. For standard Bayesian Optimization ($M = 1$), we employ the Expected Improvement (EI) acquisition function (Mockus, 1998; Jones et al., 1998; Jones, 2001). EI selects the next mixture $\boldsymbol{\pi}_{t+1}$ point by computing the expected improvement over the current best observed function value $y^* := \max\{y_1, y_2, \ldots, y_t\}$:

$$\alpha_t^{\mathtt{EI}}(\boldsymbol{\pi}) = \mathbb{E}\left[\max(0, y_{t+1} - y^*) | \boldsymbol{\pi}_{t+1} = \boldsymbol{\pi}\right] \tag{6}$$

where the expectation is taken over the predictive distribution $y_{t+1}$ given by the Gaussian process posterior mean and variance at $\boldsymbol{\pi} = \boldsymbol{\pi}^{t+1}$. At each iteration, the GP model is fitted to the accumulated observations $\mathcal{D}_{1:t}$, and the next query point $\boldsymbol{\pi}_{t+1}$ is determined by searching the input space for the point with highest improvement using gradient ascent

$$\boldsymbol{\pi}_{t+1} = \arg\max_{\boldsymbol{\pi} \in \Pi} \alpha_t^{\mathtt{EI}}(\boldsymbol{\pi}). \tag{7}$$

A new training dataset is constructed according to the selected mixture, the proxy model is trained, and the validation score $y_{t+1}$ is measured. The new data point $(\boldsymbol{\pi}_{t+1}, y_{t+1})$ is incorporated into the dataset $\mathcal{D}$ for subsequent iterations.

In MFBO, multiple proxy models $M > 1$ are available, allowing the algorithm to jointly select both a mixture and model size $(\boldsymbol{\pi}, m)$ at each iteration. The model size serves as a fidelity indicator, where larger models provide more accurate validation scores at higher computational cost, while smaller models offer less precise estimates at reduced cost. This framework enables automatic cost-accuracy trade-offs by leveraging correlations between different fidelity levels.

For MFBO, we adopt the Max-value Entropy Search (MES) acquisition function (Wang & Jegelka, 2017a). Unlike earlier entropy-based methods that focus on reducing uncertainty about the location of the optimum, MES aims to minimize uncertainty about the *maximum value $y_* = f(\boldsymbol{\pi}_*, M)$*. Mathematically, MES quantifies

the expected reduction in entropy $\mathcal{H}$, or equivalently the mutual information $\mathcal{I}$, between the maximum $y_*$ and the next observation $y_{t+1}$:

$$\alpha_t^{\texttt{MES}}(\boldsymbol{\pi}, m) = \frac{1}{c_m} \mathcal{I}(y_{t+1}, y_* \mid \mathcal{D}_{1:t}, (\boldsymbol{\pi}, m)_{t+1} = (\boldsymbol{\pi}, m)) \tag{8}$$

$$= \frac{1}{c_m} \mathcal{H}(y_{t+1}) - \mathbb{E}_{y_*}[\mathcal{H}(y_{t+1} \mid y_*)] \tag{9}$$

$$\approx \frac{1}{c_m} \frac{1}{N} \sum_{y_* \in Y_*} \left[ \frac{\gamma_{y_*}(\boldsymbol{x}) \psi(\gamma_{y_*}(\boldsymbol{x}))}{2\Psi(\gamma_{y_*}(\boldsymbol{x}))} - \log(\Psi(\gamma_{y_*}(\boldsymbol{x}))) \right] \tag{10}$$

where $\psi$ and $\Psi$ denote the probability density and cumulative distribution functions of the standard normal distribution, respectively, and $\gamma_{y_*}(\boldsymbol{x}) = \frac{y_* - \mu_t(\boldsymbol{x})}{\sigma_t(\boldsymbol{x})}$. The expectation in Equation 9 is taken over $p(y_*|D_n)$, which is approximated via Monte Carlo sampling of $N$ function maxima. The cost normalization factor $\frac{1}{c_m}$ balances information gain against computational expense, favouring cheaper low-fidelity evaluations when their information content justifies the cost reduction. At a given iteration, the next query mixture-model point $\boldsymbol{\pi}_{t+1}, m_{t+1}$ is determined by jointly optimizing over the mixture and model space: $(\boldsymbol{\pi}, m) \in \Pi \times \mathcal{M}$,

$$(\boldsymbol{\pi}_{t+1}, m_{t+1}) = \arg\max_{\boldsymbol{\pi}, m} \alpha_t^{\texttt{MES}}(\boldsymbol{\pi}, m). \tag{11}$$

This formulation enables the algorithm to automatically determines which proxy model $m$ to train next, enabling automatic balance of the exploration-exploitation trade-off across both the mixture space and fidelity levels.

### 4.3 Recommending a Mixture for the Target Model

At any iteration $t$, or upon exhausting the computational budget $C$, both standard BO and MFBO require selection of an optimal mixture $\boldsymbol{\pi}_r$ for training the target model. We leverage the posterior mean of the GP model fitted to all collected observations so far to identify the optimal mixture with the highest predicted performance for the target model $\tilde{m}$, that is

$$\boldsymbol{\pi}_r = \arg\max_{\boldsymbol{\pi} \in \Pi} \mu(\boldsymbol{\pi}, \tilde{m}). \tag{12}$$

## 5 An Instruction Finetuning Dataset for Data Mixing

A possible hurdle for the research advocated thus far is that developing new methods requires many iterations of model design and experimentation, making it prohibitively expensive for all but a few labs to contribute. To widen participation in this research field, we therefore construct and release an open dataset *ADMIRE IFT Runs* containing full fine-tuning and evaluation runs for 460 state-of-the-art LLMs (building on the Qwen2.5 base model family (Yang et al., 2024a)) across three model sizes (500m, 3b, 7b) on 256 diverse data mixtures, using realistic post-training pipelines and including results on 17 standard benchmarks following the open Tülu 3 post-training recipes of (Lambert et al., 2024). Overall, *ADMIRE IFT Runs* was constructed for a total of **13,119 GPU hours** on `nvidia-a100-80gb` GPUs. In addition to providing a realistic and diverse set of datasets to compute the mixture over, we include both development (in-distribution, ID) and unseen (out-of-distribution, OOD) datasets. As LLM post-training often directly targets benchmark results (with individual datasets specifically designed to increase results), including OOD evaluations allows us to study the effects of data mixing and training on a more meaningful level of evaluation. Using the data provided, research on regression-based data mixing techniques can essentially be carried out without running any actual costly LLM training, broadening access to research.

### 5.1 Uncovering the complex data-mixture / performance relationship

The direct analysis of the dataset reveals an intriguing structure, speaking to the complexity of developing a genuine understanding of data mixture effects on final learning results. First, consider 3a. Shown is the

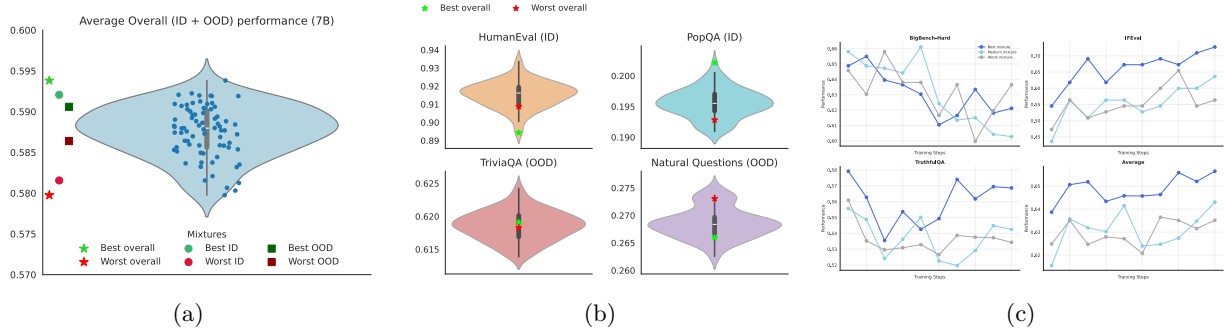

Figure 3: *ADMIRE IFT Runs* dataset exploration for 7b model runs. (a) Overall distribution of final average validation performance across in-distribution (ID) and out-of-distribution (OOD) datasets. Shown are also (b) Performance of the worst and best overall mixtures on selected sub-domains. Optimisation on average scores is sensitive to significant underperformance on individual domains. (c) Validation performance (computed on a 10% subset of the validation data) as a function of training progress.

average overall (across in-distribution and out-of-distribution datasets) performance for all 7b IFT training runs. Coloured markers furthermore show the performance of the best and worst mixtures chosen by varying criteria. By design, if mixtures are chosen according to the target metric, they correspond to the top and bottom of the distribution, respectively (stars). In practice, it is common not to run additional OOD evaluations and simply choose the best mixture according to our standard ID Evaluations (circles). However, results show that this is not advisable, as the chosen mixtures underperform when taking into account OOD tasks. On the other extreme, one might optimize for only OOD tasks to directly target challenging evaluation settings. This, too, can lead to poor results (squares), providing evidence for the argument that meaningful evaluation must be performed over the full spectrum of (ID+OOD) tasks.

The relationship between the overall best model on average and the best models per task is more complex than is often assumed, as shown in Figure 3a. Despite the chosen mixture (green star) leading to the best overall results (across ID+OOD) we find that results on individual evaluation sets show performance near the top, median or even at the very bottom of the distribution (`HumanEval`). More surprisingly, in several instances, the worst overall model outperforms the best overall model by a significant margin. For researchers particularly concerned about certain evaluations (e.g. safety/ethics datasets), more robust metrics than the simple arithmetic mean may be more suitable and may be studied with the help of the provided dataset.

Finally, consider the validation performance[2] as a function of training steps. Far from monotonic increases on all domains, we observe evidence of catastrophic forgetting (French, 1999; Kirkpatrick et al., 2017; Schwarz et al., 2018) experienced during the IFT training stage (relative to the base model), see `BigBench-Hard` and `TruthfulQA`. However, provided the mixture is carefully chosen such forgetting can be reduced (see the best mixture curve on `TruthfulQA`), presumably by including sufficient data similar to the skills being tested. This can be seen as an instance of rehearsal-based Continual Learning (Rolnick et al., 2019), automatically being discovered by data-mixture optimization. Full results for 3b and 3c across all domains and model sizes can be found in the Appendix.

## 5.2 Interpretable Relationship Between Training Mixture Weights and Target Domain Performance

In addition, *ADMIRE IFT Runs* allows an analysis of the impact of training datasets on evaluation (both ID & OOD) results. In particular, the kernel matrices in Figure 4 represent dataset importance scores obtained through a GP using an Automatic Relevance Determination (ARD) kernel. This kernel models the relationship between source domain weights in training data mixture and target domain performance using a zero-mean Gaussian Process emulator with Maximum-Likelihood estimation of hyperparameters. Each hyperparameter in the ARD kernel represents characteristic length scales that can be interpreted as

---

[2]computed on 10% of the validation dataset for efficiency reasons

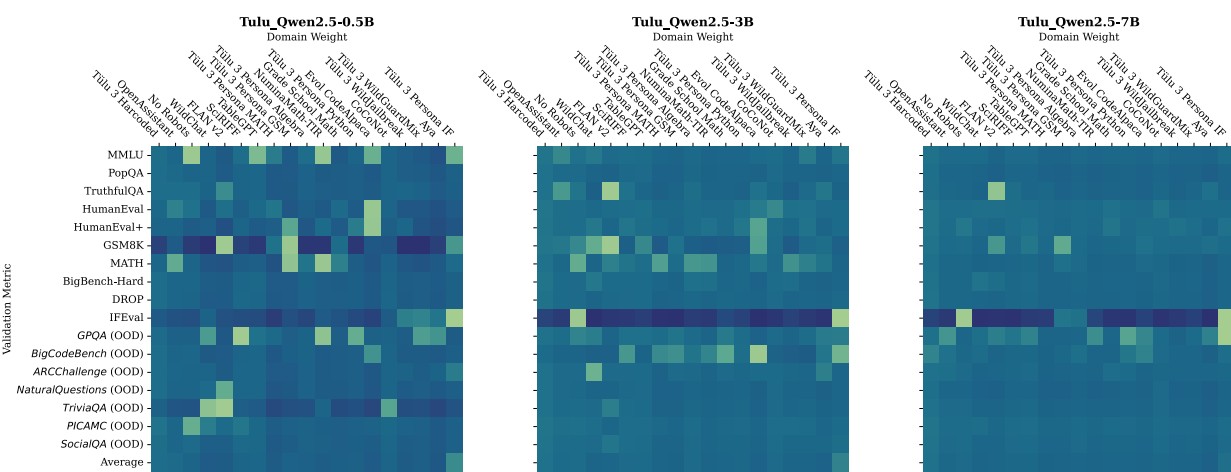

Figure 4: Estimated importance of each source domain to each evaluation benchmark for the Tülu 3 evaluation suite using a Gaussian Processes with Automatic Relevance Determination (ARD) Kernel. Estimated importance metrics show a stronger transfer between Qwen2.5-3B and Qwen2.5-7B. Lighter/Darker colors correspond to higher/lower importance. Best viewed on a computer.

sensitivity measures, where smaller length scales indicate higher influence of the corresponding particular input dimensions.

The estimated importance matrices reveal meaningful and interpretable source-target domain relationships that validate the effectiveness of our approach. For instance, as expected, `GSM8K` demonstrates strong correlations with mathematics-focused training datasets such as `Tülu 3 Persona Math` and `Tülu 3 Persona GSM`, as well as reasoning-heavy data included in the `FLAN v2` collection, which aligns with the mathematical reasoning requirements of the `GSM8K` benchmark. Similarly, `IFEval` exhibits pronounced influence from `Tülu 3 Persona IF` (a dataset designed to improve `IFEval`) and `No Robots` training datasets. Upon closer inspection, we discovered that numerous training examples in the `No Robots` dataset contain strict instruction-following patterns, explaining this strong correlation. These results demonstrate that our method yields insightful source-domain relationships for curating meaningful training data mixtures while providing crucial interpretability for understanding model behaviour.

Another interesting finding is a significant reduction in the number of important training domains as model scale increases from 0.5B to 7B parameters, which is particularly evident when comparing the density of lighter colors (indicating higher importance) across the three heatmaps. We hypothesize that larger models inherently possess greater amounts of pre-trained knowledge and capabilities compared to smaller counterparts (an argument/observation frequently made by post-training teams (Abdin et al., 2024, e.g.)). On the other hand, the importance matrix for Qwen2.5-3B is, for the most part, fairly similar to the larger model counterpart, revealing opportunities for transferable insights at smaller scale.

Consequently, downstream performance on common evaluation benchmarks may become less dependent on external training factors. This hypothesis is further supported by the relatively smaller performance gap between best and worst data mixture configurations observed in larger LLMs compared to the smallest model size (in Figures 9-11, Appendix), suggesting that larger models may be more robust to variations in training data composition due to their enhanced inherent capabilities.

### 5.2.1 Dataset contstruction

We select the Tülu-3-SFT mixture from Lambert et al. as our foundation for data mixture creation. This publicly accessible post-training dataset contains 939,344 samples spanning 17 datasets across diverse domains: mathematics, coding, reasoning, instruction-following, knowledge, and safety. Taking inspiration from Liu et al. (2024), we sample 256 distinct dataset mixtures that comprehensively cover the probability space of the

17-dimensional simplex corresponding to the datasets in the Tülu-3-SFT mixture. We achieve this coverage using a Dirichlet distribution parameterized by the optimal dataset weights from the original Tülu-3-SFT mixture as priors—weights that were extensively optimized through human expertise and iterative refinement. This prior-based approach ensures our sampled mixtures statistically reflect realistic data availability patterns while enabling exploration of both sparse and near-uniform distributions.

**Post-training Setup** To maintain practical relevance and avoid overfitting to specific SFT data mixtures, we limit each data mixture to 200,000 training samples. We train Qwen2.5-0.5B models on all 256 mixtures, Qwen2.5-3B models on 128 mixtures, and Qwen2.5-7B models on 76 randomly selected mixtures from the 3B subset. All post-training experiments strictly adhere to the open-instruct training pipeline and hyperparameters established in the original Tülu 3 project.[3]

**Evaluation Protocol** Following the established evaluation protocol from the original Tülu 3 work, we evaluate trained models on two distinct benchmark suites. For in-distribution (ID) evaluation, we employ the Tülu 3 development set, which comprises 12 carefully selected LLM evaluation benchmarks with rigorous decontamination against training data. These ID results serve as the targets for our data mixture optimization.

**Dataset Contributions** The resultant *ADMIRE IFT Runs* dataset represents a significant contribution to the research community, providing public access to 460 trained checkpoints across 256 diverse data mixtures, accompanied by comprehensive evaluation results on 17 standard benchmarks. We demonstrate initial applications through several case studies examining data mixture optimization, zero-shot transferability analysis, and multi-fidelity scaling studies, while anticipating that future research will uncover additional applications we have not yet explored.

## 6 Experiments

**The Pile Mixture Dataset** Apart from the *ADMIRE IFT Runs* explained in section 5, we conduct experiments on open-sourced benchmark datasets from RegMix, which comprises $256\times$ pre-training and evaluation results across three different model scales (1M, 60M and 1B parameters) on the Pile dataset (Gao et al., 2020). Each data point in the benchmark consists of input features representing the proportions of various training domains from the Pile, with corresponding evaluation performance (measured in perplexity) of the trained models on evaluation domains that form a subset of the training domains.

**Baselines** In our framework, a data mixture optimization algorithm consists of a method to sequentially propose new mixtures and models $(\boldsymbol{\pi}_{t+1}, m_{t+1})$ and a method to recommend a final best single mixture $\boldsymbol{\pi}_r$. We compare our proposed method against strong baselines that fit parameterized models to observations of evaluation metrics as functions of training data mixtures, then leverage these models to recommend a final data mixture. However, these (non sequential) baselines do not explicitly define an iterative data collection method. We therefore adapt RegMix that randomly sampled 512 data mixtures. At each iteration, for a pre-specified constant model size $m$, a single random mixture is chosen from the benchmark dataset, $\boldsymbol{\pi}_{t+1}$, and we look up its corresponding validation score $y_{t+1}$. These baseline methods typically require prior knowledge of downstream evaluation tasks, such as access to a small validation dataset.

- **RegMix** (Liu et al., 2024): fits a linear regression model on existing observations with weights $w_i$ and intercept $w_0$

$$y = \sum_{i=1}^{d} w_i \pi_i + w_0,  \tag{13}$$

  to predict the best data mixture. Note, the subscript $i$ denotes elements of the vector $\boldsymbol{\pi}$.

- **Data Mixing Law (DML)** (Ye et al., 2024): fits an exponential regression model

$$y = \theta \exp\left(\sum_{i=1}^{n} w_i \pi_i\right) + w_0,  \tag{14}$$

---

[3]https://github.com/allenai/open-instruct

where $\theta$ is also a learned parameter.

- **MFMS-GP** (Yen et al., 2025): a concurrent method that leverages multi-fidelity, multi-scale Bayesian optimization to construct a parameterized performance predictor, enabling optimal selection of data mixtures for training.

- **Support Vector Machine (SVM)** (Fan et al., 2008; Chang & Lin, 2011; Bishop & Nasrabadi, 2006; Smola & Schölkopf, 2004): fits a linear Support Vector Regression model.

We also we consider baselines that do not rely on any prior knowledge of downstream evaluation for data mixture optimization:

- **Random Selection**: Randomly selects a data mixture with uniform probability from the search space as the recommended optimal mixture, serving as a lower-bound for evaluating algorithm performance.

- **DoReMi** (Xie et al., 2023): trains a small proxy model using group distributionally robust optimization over training datasets to produce data mixture weights (mixture proportions).

**Implementation**  For ADMIRE-BayesOpt and all baseline approaches, we optimize the training data mixture over steps until a predefined maximum acquisition budget $C$ is exhausted. The cost associated with each optimization step in the *ADMIRE IFT Runs* corresponds to the average wall-clock training time for each model size. Due to the unavailability of actual training times in the RegMix paper, we assume cost scales linearly with model size and use model size as a proxy for cost in our quantitative analysis.

For zero-shot transfer experiments (single-fidelity BO), all methods are constrained to acquire and observe data points from a single fidelity level (proxy model size) during data mixture optimization. In contrast, for MFBO, methods can query and observe data points across all proxy model sizes.

We implement our method using the popular open-source Bayesian optimization library BoTorch (Balandat et al., 2020). At each optimization step, all methods recommend their best-performing data mixture within the candidate set of the target model (1B model for the Pile and 7B model for the *ADMIRE IFT Runs*) based on acquired training data points. We evaluate these methods by comparing the performance of the target model trained on their respective recommended data mixtures. Throughout the optimization process, we report cumulative best performance metrics to demonstrate both the efficiency and effectiveness of our Bayesian optimization approach. All reported results are averaged over 5 independent runs.

### 6.1 Zero-Shot Transfer Results

We first compare ADMIRE-BayesOpt to all baselines when choosing data mixtures for smaller models, then evaluating the mixtures by training and evaluating bigger models. This is the same setup as they propose, with the only addition being sequential choice of mixture, allowing a head-to-head comparison. In subsection 6.2, we add multiple model sizes to examine data mixers that can also choose model sizes. Our main results are shown in Figure 5, and we discuss results for each dataset below.

#### 6.1.1 ADMIRE-BayesOpt Demonstrates Superior Transferability Across Model Scales

**The Pile Dataset Performance**  As shown in Figure 5a, ADMIRE-BayesOpt consistently achieves optimal data mixture identification across all model scales, demonstrating robust transferability from 1M to 1B parameter models. We also observe that ADMIRE-BayesOpt and RegMix are the only methods capable of identifying optimal mixtures for The Pile dataset, while baseline methods (DML, SVM, Random Selection) fail to converge within allocated compute budget.

In extreme transfer scenarios (1M→1B parameters), ADMIRE-BayesOpt is the *only* method successfully identifying the optimal mixture for 1B model, while RegMix requires full computational budget yet achieves 0.3% lower validation performance. This transferability spans three orders of magnitude in model size, suggesting that ADMIRE-BayesOpt effectively captures mixture-performance relationships that remain consistent across architectural scales.

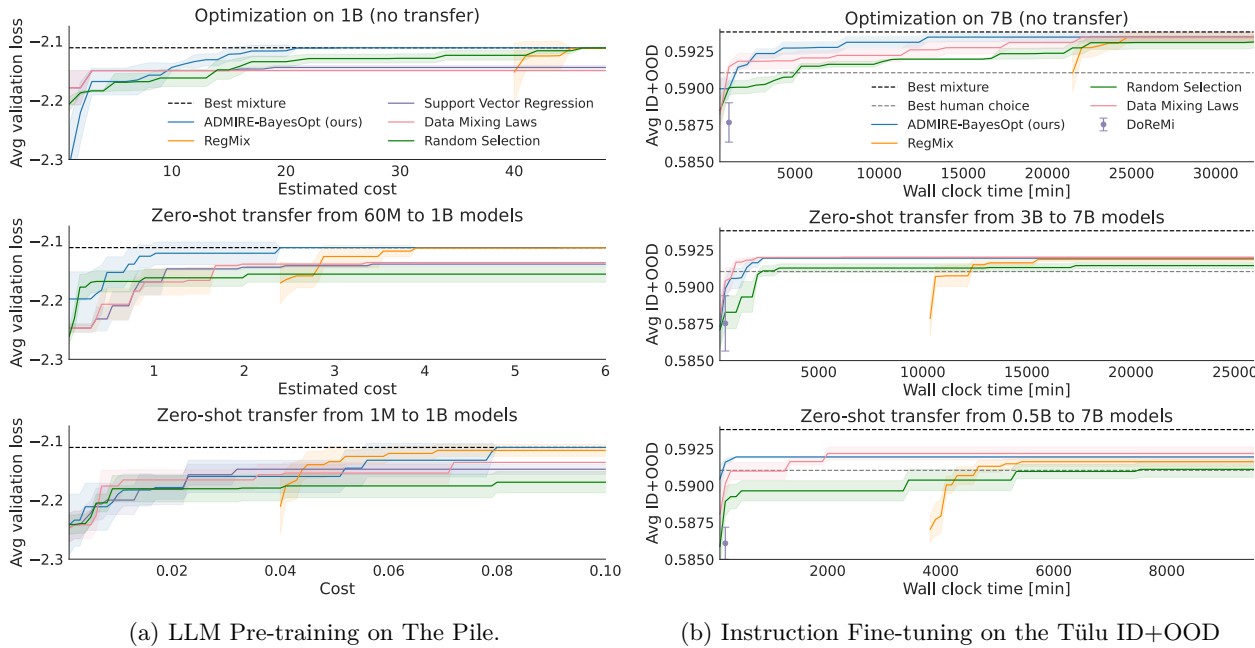

(a) LLM Pre-training on The Pile.  (b) Instruction Fine-tuning on the Tülu ID+OOD

Figure 5: Zero-shot transfer performance comparison across model scales, showing performance when optimizing on smaller proxy models and transferring to the larger target model.

**The *ADMIRE IFT Runs* Dataset Performance**  As shown in Figure 5b, both ADMIRE-BayesOpt and RegMix successfully identify mixtures superior to human-optimized baselines across all transfer scenarios (0.5B→7B, 3B→7B) on the Tülu dataset, while other baselines (DML, SVM, Random Selection) fail entirely. Notably, DoReMi performs worse than the original Tülu mixture, indicating that not all optimization approaches can improve upon human expertise.

### 6.1.2 ADMIRE-BayesOpt Achieves Significant Computational Efficiency Gains

In Figure 5a, ADMIRE-BayesOpt demonstrates superior efficiency compared to all baseline methods on the Pile dataset, requiring significantly lower computational cost to identify optimal data mixtures. When training and evaluating on 1B model data, ADMIRE-BayesOpt achieves the optimal mixture at $1.86\times$ lower cost than Random Selection, which eventually discovers the same optimal mixture but at substantially higher expense. In zero-shot transfer scenarios, ADMIRE-BayesOpt's efficiency advantages become even more pronounced: transferring from 60M to 1B models, ADMIRE-BayesOpt finds the best mixture with $2.36\times$ lower cost compared to RegMix, while other baselines fail to identify the optimal mixture.

On the *ADMIRE IFT Runs* in Figure 5b, ADMIRE-BayesOpt consistently outperforms RegMix across all evaluation domains with substantial speed improvements. For the OOD+ID domain, ADMIRE-BayesOpt achieves a $2\times$ speed-up over RegMix when recommending the highest mixture using 7B model data, and demonstrates $19\times$ faster performance in the 0.5B to 7B transfer setting. In OOD domains specifically, ADMIRE-BayesOpt shows even greater efficiency gains with a $7.14\times$ speed-up over RegMix for 7B model data and $17.5\times$ faster performance in the 0.5B to 7B transfer scenario. For ID domains, ADMIRE-BayesOpt maintains its efficiency advantage with a $2.87\times$ speed-up over RegMix, and continues to demonstrate $2\text{-}2.25\times$ cost improvements across different transfer settings.

### 6.1.3 Increased Transferability with Proxy Model Scale

Our experimental evaluation reveals critical insights into the transferability challenges of data mixture optimization methods, particularly in zero-shot transfer scenarios. On the *ADMIRE IFT Runs* in Figure 5b, we observe a consistent pattern where transferability— in terms of converged final performance —improves

substantially with increasing proxy model sizes. For example, when transferring the discovered optimal data mixture from smaller to larger models, ADMIRE-BayesOpt progressively achieves better performance: reaching final performances of 59.19→59.35 for 0.5→7B proxy models.

These findings expose a fundamental limitation in existing data optimization approaches: their inability to effectively transfer knowledge from significantly smaller proxy models alone. These observations support our motivation to incorporate multi-fidelity Bayesian optimization techniques into our method, which allows ADMIRE-BayesOpt to effectively capture the complex relationships between data mixtures and performance across all model scales and improve convergence and efficiency in data mixture optimization.

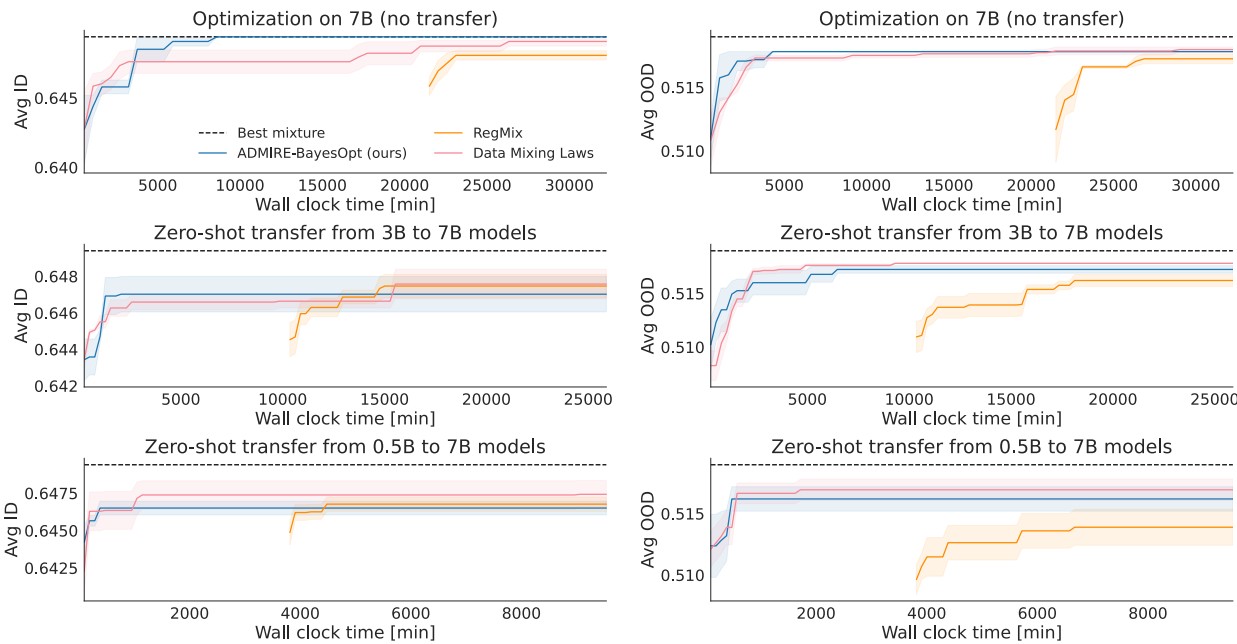

Figure 6: Zero-shot transfer performance on Tülu 3 instruction fine-tuning across different evaluation domains. *Left*: Average performance on In-Distribution (ID) evaluation datasets; *Right*: Average performance on Out-of-Distribution (OOD) evaluation datasets

## 6.2 Multi-Fidelity Results

We next consider the multi-fidelity setting, where a data mixing solution sequentially picks both a data mixture *and* a model size to train on it. To the best of our knowledge, this is unconsidered by prior works. Our results are shown in Figure 7, where we overall see that find that ADMIRE-MFBO approach demonstrates substantial improvements in both efficiency and efficacy compared to our single-fidelity ADMIRE-BayesOpt and baseline approaches. In Figure 7a, the Pile dataset, ADMIRE-MFBO achieves the optimal data mixture using only 7.73 cost units, representing an 82.82% cost reduction compared to the Pareto Frontier of baseline methods optimizing directly on the target model (which required 45 cost units) and a 67.79% improvement over our single-fidelity ADMIRE-BayesOpt approach (which required 24 cost units). Similarly, In Figure 7b, the *ADMIRE IFT Runs*, ADMIRE-MFBO identifies the optimal mixture in just 5651.68 minutes, delivering 2.19× and 4.37× speed improvements over ADMIRE-BayesOpt and baseline methods, respectively, while being the only algorithm to successfully find the true optimal mixture.

The superior performance of ADMIRE-MFBO stems from its intelligent integration of information across multiple fidelity levels, particularly through strategic leveraging of low-fidelity data in early optimization stages. As illustrated in Figure 2b, ADMIRE-MFBO adopts a progressive sampling strategy that begins with almost exclusive use of the lowest fidelity data (low-fidelity dominance) before gradually transitioning to more costly evaluations, culminating with the highest fidelity data sampling (multi-fidelity integration). This multi-fidelity approach creates the characteristic "step function" optimization curves, where the algorithm initially relies on

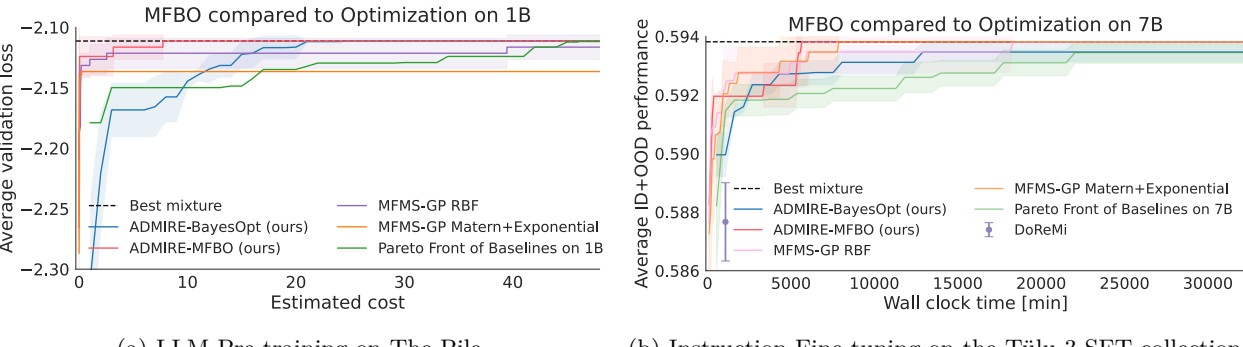

(a) LLM Pre-training on The Pile.

(b) Instruction Fine-tuning on the Tülu 3 SFT collection.

Figure 7: Multi-Fidelity Bayesian Optimization (MFBO) results. The line in green shows a unified, Pareto-optimal baseline constructed from the element-wise best performance across all single-fidelity methods at each time step.

inexpensive low-fidelity evaluations that provide suboptimal but low-cost mixture recommendations during approximately 12% of the optimization process. Subsequently, ADMIRE-MFBO transfers relevant knowledge from lower to higher fidelities through a critical period of combining all-level information, ultimately enabling rapid convergence to optimal solutions while maintaining both superior effectiveness and efficiency compared to single-fidelity approaches like the bassline and ADMIRE-BayesOpt.

## 6.3 Additional Analysis

### 6.3.1 Per-domain Evaluation Performance of Data Mixtures

We conduct an additional detailed analysis on the data mixture optimization process by examining the per-domain evaluation metrics across all candidate data mixtures from the *ADMIRE IFT Runs*. The violin plots for the 0.5B, 3B, and 7B models (Figures 9-11, Appendix) reveal substantial heterogeneity in domain sensitivity to data mixture ratios. As evidenced by the violin plots for the 0.5B, 3B, and 7B models, certain domains—such as `IFEval` and `HumanEval`—exhibit pronounced variability in performance across mixtures, indicating a high degree of sensitivity to data composition. In contrast, domains like `MATH` and `GSK8K` display narrower distributions, suggesting robustness to changes in the corresponding data mixture ratios. Notably, the degree of sensitivity is not uniform across model scales; for example, `HumanEval` shows considerable spread in the 0.5B model but becomes markedly less sensitive as model size increases, whereas domains such as `TruthfulQA` retain sensitivity across all scales.

An important caveat emerges when optimizing the training data mixture solely for a single overall metric—defined here as the unweighted mean across all domains. While this approach consistently yields the highest aggregate performance, it can inadvertently mask suboptimal or even degraded performance in individual domains, as illustrated by the relative positions of the best-overall (green) and worst-overall (red) mixtures in the plots. This phenomenon underscores a key limitation of single-metric optimization and motivates further research into alternative mixture optimization objectives, such as multi-task or domain-weighted optimization, to better balance per-domain performance and mitigate potential failure cases in critical domains.

### 6.3.2 Computational Cost

While Bayesian optimization introduces additional computational steps compared to simpler heuristics, we demonstrate that these costs are negligible in the context of data mixture optimization. In Table 1, we profiled the wall-clock time of each component across 100 optimization iterations. GP posterior updates require an average of 2.95 seconds per iteration, while acquisition function optimization takes 0.043 seconds. In contrast, training and evaluating a 7B language model on a candidate mixture requires approximately 12

Table 1: Computational cost breakdown per optimization iteration

| Component | Wall-Clock Time | Percentage of Total Cost |
|---|---|---|
| *ADMIRE-BayesOpt Components* | | |
| GP Posterior Update | 2.95 sec | <0.01% |
| Acquisition Optimization | 0.043 sec | <0.01% |
| LLM Training & Evaluation | ~12 hours (7B) | >99.9% |

hours. Thus, all BO-related computations account for less than 0.01% of the total cost per iteration, with LLM training dominating at over 99.9%.

This cost structure highlights the core value proposition of ADMIRE-BayesOpt: since LLM training is unavoidable and overwhelmingly expensive across all data mixture optimization methods, even modest reductions in the number of required training runs yield substantial practical benefits. Our method achieves precisely this through principled sequential decision-making, resulting in the significant speedups reported in Figures 5-7. The minimal overhead of Bayesian optimization is therefore highly justified by the dramatic reduction in expensive model training iterations.

### 6.3.3 Ablation Study: Kernel and Acquisition Function Design Choices

**Model Kernel Ablation** To justify our Downsampling kernel design for modeling correlations across model scales, we compare alternative kernel formulations on the *ADMIRE IFT Runs* (Qwen2.5-7B target model). All experiments use identical settings, varying only the model kernel $\mathbf{K}^{\text{DS}}(m, m')$ while maintaining the RBF kernel in $\mathbf{K}^{\text{MFBO}}$.

Table 2: Ablation study on model kernel choices for ADMIRE-MFBO

| Kernel Configuration | Best Performance ↑ | Time to Best (min) ↓ |
|---|---|---|
| RBF+Downsampling (Ours) | **0.594** | **5651** |
| RBF+RBF | **0.594** | 17500 |
| Matérn+Downsampling | 0.592 | 6365 |

In Table 2, our RBF+Downsampling kernel achieves the best final performance while requiring substantially less wall-clock time. Replacing the Downsampling kernel with a symmetric RBF kernel (RBF+RBF) matches final performance but incurs a $\sim 3\times$ increase in computational cost. The Matérn+Downsampling variant demonstrates robustness to mixture-space kernel choice, though it slightly underperforms in both accuracy (0.592 vs. 0.594) and efficiency.

Beyond these empirical comparisons, the effectiveness of our simple parameter-based model kernel warrants further explanation: Our parameter-based Downsampling kernel is specifically designed to capture correlation in *performance trends* rather than absolute performance values between models of different scales. This design encodes a well-motivated prior: models with similar parameter counts, when trained on the same data mixture, exhibit correlated directional improvements—when one model's performance increases with a particular mixture, similarly-sized models tend to show comparable trends, even if their absolute performance differs substantially. This simplified kernel choice is particularly well-suited to data mixture optimization, where: (1) practitioners typically employ architecturally similar models from the same family (Xie et al., 2023; Ye et al., 2024), and (2) the extreme low-data regime (recall that our 460 training runs require 13,000 A100 GPU hours) makes simpler kernels with strong inductive biases preferable to sophisticated alternatives like neural network kernels Wilson et al. (2016), which risk overfitting without sufficient observations. The empirical validation in Section 6.1.3 further supports this design: ADMIRE-BayesOpt achieves progressively better transferability as proxy model size approaches target size (59.19→59.35 for 0.5B→7B), confirming that models closer in scale indeed exhibit higher performance correlation.

**Acquisition Function Ablation**   We evaluate acquisition function choices in both single-fidelity and multi-fidelity settings. For ADMIRE-BayesOpt, we compare Expected Improvement (EI) against Upper Confidence Bound (UCB) and Probability of Improvement (PI) in the no-transfer setting, optimizing directly on the 7B target model. For ADMIRE-MFBO, we compare MF-MES against multi-fidelity variants (CostAwarePI, CostAwareUCB). All experiments report results up to 30,000 minutes of wall-clock time, with the optimal data mixture achieving 0.594 as the performance upper bound.

Table 3: Ablation study on acquisition function choices

| *ADMIRE-BayesOpt (Single-Fidelity)* | | |
|---|---|---|
| **Acquisition Function** | **Best Performance** ↑ | **Time to Best (min)** ↓ |
| EI (Ours) | **0.593** | 5912 |
| PI | 0.591 | **537** |
| UCB | 0.590 | **537** |
| *ADMIRE-MFBO (Multi-Fidelity)* | | |
| **Acquisition Function** | **Best Performance** ↑ | **Time to Best (min)** ↓ |
| MF-MES (Ours) | **0.594** | 5651 |
| CostAwarePI | **0.594** | 26293 |
| CostAwareUCB | 0.592 | **953** |

As shown in Table 3, in the single-fidelity setting, EI achieves the best final performance (0.593), outperforming both PI and UCB. While PI and UCB converge faster initially, they plateau at suboptimal solutions. In the multi-fidelity setting, MF-MES successfully identifies the optimal data mixture (0.594), while CostAwareUCB plateaus at 0.592. Although CostAwarePI eventually matches the optimal performance, it requires $\sim 5\times$ more wall-clock time than MF-MES. These results suggest that MF-MES's information-theoretic approach to balancing exploration and exploitation across fidelities is most effective for data mixture optimization, justifying our design choices.

## 7   Discussion and Limitations

While ADMIRE-BayesOpt and ADMIRE-MFBO demonstrate substantial improvements in data mixture optimization efficiency and effectiveness, we acknowledge a few important limitations that point toward valuable directions for future research. We hope that explicitly documenting these limitations and potential solutions will provide the community with concrete pathways for extending this line of work to increasingly complex and realistic data-mixture optimization scenarios.

**Kernel Design**   Our current approach employs a parameter-count-based downsampling kernel to model correlations across model scales, which provides sufficient performance in the low-data regime characteristic of practical data mixture optimization. While this design is motivated by well-established literature (Wu et al., 2020; Kaplan et al., 2020) and validated by our transferability results, extending our framework with more sophisticated kernel designs represents a promising direction for future work. Potential approaches include developing kernels inspired by deep kernel learning (Wilson et al., 2016) that could capture more complex performance relationships across models. Such extensions, while requiring more observations to train effectively, could further enhance the flexibility and applicability of our framework.

**Extensions to Dynamic Environments**   Our method currently assumes a fixed training and evaluation environment, which is consistent with existing data mixture optimization approaches such as DoReMi (Xie et al., 2023) and Data Mixing Laws (Ye et al., 2024). While this reflects common practice in the field, we acknowledge that it limits direct applicability to more dynamic scenarios involving continual learning or evolving data sources. Importantly, our framework can be adapted to accommodate evolving data sources without complete re-optimization. When a new domain becomes available, the optimization can continue by expanding the mixture weight dimensionality, treating all previous observations as valid configurations with

zero weight on the new domain. The GP's inherent uncertainty quantification will then guide targeted exploration of mixtures involving the new domain through the acquisition function. More sophisticated dynamic environments could potentially be addressed through online Bayesian optimization strategies (Bogunovic et al., 2016) or meta-learning approaches for rapid adaptation to distribution shifts (Wei et al., 2021).

**Theoretical Analysis and Convergence Guarantees**    While our work demonstrates empirical improvements in data mixture optimization efficiency, we acknowledge the absence of formal theoretical analysis specific to our multi-fidelity setting. For standard Bayesian optimization with classical acquisition functions like Expected Improvement, well-established regret bounds and convergence guarantees from the literature directly apply to ADMIRE-BayesOpt. However, deriving comprehensive regret bounds for multi-fidelity Max-value Entropy Search remains, to the best of our knowledge, an active area of research within the Bayesian optimization community. While foundational work exists for single-fidelity MES (Wang & Jegelka, 2017b) and certain multi-fidelity settings (Song et al., 2019; Zhang et al., 2025), complete theoretical characterization of the continuous multi-fidelity MES framework we employ represents an open problem. Extending these theoretical guarantees to our setting would strengthen the foundations of data mixture optimization methods and represents an important direction for future work, both for the broader Bayesian optimization community and for our specific application domain.

## 8    Conclusion

We propose ADMIRE-BayesOpt, a Bayesian Optimization framework designed to improve the efficiency and effectiveness of discovering optimal data mixtures. Our standard BO method demonstrates strong performance on single-fidelity data and exhibits robust transferability to higher-fidelity settings. The Multi-Fidelity Bayesian Optimization (MFBO) variant further enhances efficiency by strategically allocating expensive high-fidelity evaluations, enabling faster convergence and lower overall computational cost. We validate the effectiveness and efficiency of ADMIRE-BayesOpt on hundreds of mixture data points from two state-of-the-art datasets: Tülu 3 and the Pile, demonstrating that our approach consistently outperforms existing baseline methods.

Our work makes several important contributions. We demonstrate that Bayesian optimization provides a principled and effective framework for data mixture optimization, achieving substantial improvements compared to recent baselines on realistic large-scale experiments. Our multi-fidelity approach introduces a natural mechanism for balancing exploration across model scales, automatically learning to leverage inexpensive low-fidelity evaluations before converging on high-fidelity configurations. Furthermore, by releasing the *ADMIRE IFT Runs* dataset containing over 13,000 GPU hours of training artifacts, we substantially lower the barrier to entry for researchers interested in this problem space, enabling future investigations without prohibitive computational requirements. This work opens the door to a wide range of future research directions, enabling more accessible exploration of sequential decision-making techniques for data mixture optimization.

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

# A   Implementation Details

To ensure consistency in implementation and enable a fair comparison between Bayesian Optimization (BO) and Multi-Fidelity Bayesian Optimization (MFBO), we follow the official BoTorch tutorial[4], which provides a unified experimental framework for both approaches.

**Bayesian Optimization**   We adopt the standard BO pipeline, in which a Gaussian Process (GP) model is iteratively fit to the observed data, and an acquisition function is optimized based on the updated model at each iteration. Specifically, we use the `SingleTaskMultiFidelityGP` model and the `qLogExpectedImprovement` acquisition function. For experiments involving a single model size, the training data contains a single fidelity level. The acquisition function is optimized using `optimize_acqf_discrete` over the training set, from which the point with the highest acquisition value is selected. We set the number of restarts to 10 and the number of raw samples to 1024.

After each GP update, the model recommends a point from the training set corresponding to the highest fidelity level (1.0). This is done by maximizing the posterior mean of the GP model while fixing the fidelity

---

[4]`https://botorch.org/docs/tutorials/discrete_multi_fidelity_bo/`

dimension. The data point with the highest posterior mean is selected. We run the optimization for a total of 100 iterations, initializing the GP model with a single observed data point.

**Multi-Fidelity Bayesian Optimization**  For MFBO, we use the same GP model to fit observations across multiple fidelity levels. Both the Tulu and the Pile datasets are composed of three fidelity levels. Specifically, the ID+OOD Tulu dataset includes 444 samples (256 at 0.5B, 128 at 3B, and 60 at 7B), while the Pile dataset contains 816 samples (512 at 1M, 256 at 60M, and 48 at 1B).

The acquisition function used is `qMultiFidelityMaxValueEntropy`, which selects the next evaluation point by performing a forward pass on the training set and choosing the one with the highest acquisition value. The same acquisition function is reused when querying the model for recommendations at the highest fidelity level. To fully capture the recommendation trajectory across fidelities, we set the total number of iterations equal to the total number of data points across all fidelity levels.

# B  Acquisition Function Optimization

The optimization of data mixtures in ADMIRE-BayesOpt involves solving a constrained optimization problem at each iteration. This section provides implementation details for both the acquisition phase (selecting the next experimental configuration) and the recommendation phase (proposing the optimal mixture for the target model).

**Constrained Optimization for Mixture Selection**  Given an acquisition function $\alpha_t(\boldsymbol{\pi}, m)$ (Expected Improvement for single-fidelity BO or Max-value Entropy Search for MFBO, as described in Section 4.2), determining the next query point $(\boldsymbol{\pi}_{t+1}, m_{t+1})$ requires solving:

$$(\boldsymbol{\pi}_{t+1}, m_{t+1}) = \underset{(\boldsymbol{\pi}, m) \in \Pi \times \mathcal{M}}{\arg \max} \alpha_t(\boldsymbol{\pi}, m) \tag{15}$$

subject to the probability simplex constraint $\sum_{i=1}^{d} \pi_i = 1$ with $\pi_i \in [0, 1]$ for all $i$. While various well-established constrained optimization methods can address this problem (Frazier, 2018; Garnett, 2023) (which are orthogonal to the main focus of our paper), we describe one implementation approach in detail below.

**Model enumeration.** Since the model space $\mathcal{M}$ is discrete and finite ($M$ models for MFBO, or $M = 1$ for standard BO), we enumerate over all candidate proxy models. For each model $m \in \mathcal{M}$, we optimize the mixture $\boldsymbol{\pi}$ as described next.

**Projected gradient ascent for mixture optimization.** For the continuous variable $\boldsymbol{\pi}$ constrained to the probability simplex, we employ projected gradient ascent. At iteration $k$, this procedure consists of two steps:

1. *Gradient ascent step*: Compute an unconstrained update

$$\tilde{\boldsymbol{\pi}}^{(k+1)} = \boldsymbol{\pi}^{(k)} + \eta \nabla_{\boldsymbol{\pi}} \alpha_t(\boldsymbol{\pi}^{(k)}, m) \tag{16}$$

   where $\eta$ is the learning rate.

2. *Simplex projection step*: Project the unconstrained iterate back onto the probability simplex

$$\boldsymbol{\pi}^{(k+1)} = \mathrm{Proj}_{\Pi}(\tilde{\boldsymbol{\pi}}^{(k+1)}). \tag{17}$$

   This can be implemented using the efficient Euclidean projection described in Duchi et al. (2008), or alternatively via the softmax operation.

**Selection of optimal query point.** After optimizing $\alpha_t(\boldsymbol{\pi}, m)$ over $\boldsymbol{\pi} \in \Pi$ for each model $m \in \mathcal{M}$, we select the model-mixture pair achieving the maximum acquisition value:

$$(\boldsymbol{\pi}_{t+1}, m_{t+1}) = \underset{m \in \mathcal{M}}{\arg \max} \max_{\boldsymbol{\pi} \in \Pi} \alpha_t(\boldsymbol{\pi}, m) \tag{18}$$

Following the selection of $(\boldsymbol{\pi}_{t+1}, m_{t+1})$, we train model $m_{t+1}$ with mixture $\boldsymbol{\pi}_{t+1}$, evaluate its performance to obtain observation $y_{t+1}$, and update the GP posterior according to Equations 4–5.

**Final Mixture Recommendation**   At any iteration $t$, or upon exhausting the computational budget $C$, we recommend an optimal mixture $\boldsymbol{\pi}_r$ for the target model $\tilde{m}$ by maximizing the GP posterior mean:

$$\boldsymbol{\pi}_r = \arg\max_{\boldsymbol{\pi} \in \Pi} \mu_t(\boldsymbol{\pi}, \tilde{m}) \tag{19}$$

This optimization employs the same projected gradient ascent procedure with simplex projection described above. Unlike the acquisition function, which balances exploration and exploitation, the posterior mean directly exploits accumulated knowledge about mixture performance.

**Implementation for Benchmark Evaluation**   As we described in Appendix A, for experimental evaluation in Section 6, we assess our method on discrete sets of mixture candidates from established benchmark datasets to enable fair comparison with prior work (Liu et al., 2024). Specifically, we evaluate the acquisition function $\alpha_t(\boldsymbol{\pi}, m)$ only at the candidate mixtures included in the benchmark, selecting the mixture with the highest acquisition value for the next query, or the highest posterior mean $\mu_t(\boldsymbol{\pi}, \tilde{m})$ for the final recommendation. We note that the continuous optimization framework described above represents the general formulation applicable when working outside benchmark constraints with discrete candidates.

## C   Additional Figures and Results

Figure 8 shows the validation performance as a function of training steps on a selected number of benchmarks from Tülu 3 for the best and worst overall mixture, as well as a mixture of medium performance.

Figures 9, 10, and 11 show the performance distribution of all trained models on the respective Tülu domains.

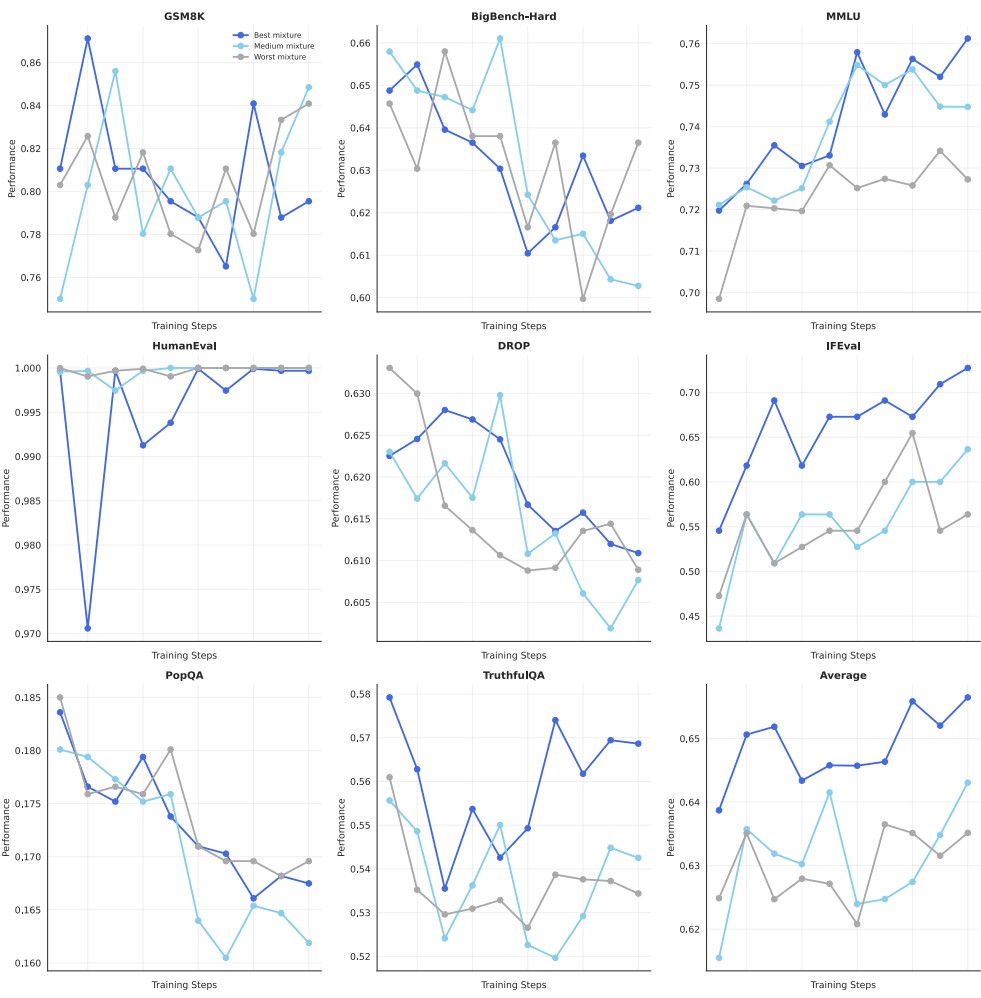

Figure 8: Validation results on various domains as a function of the number of training steps. Shown are the best, worst and a medium-performance mixture.

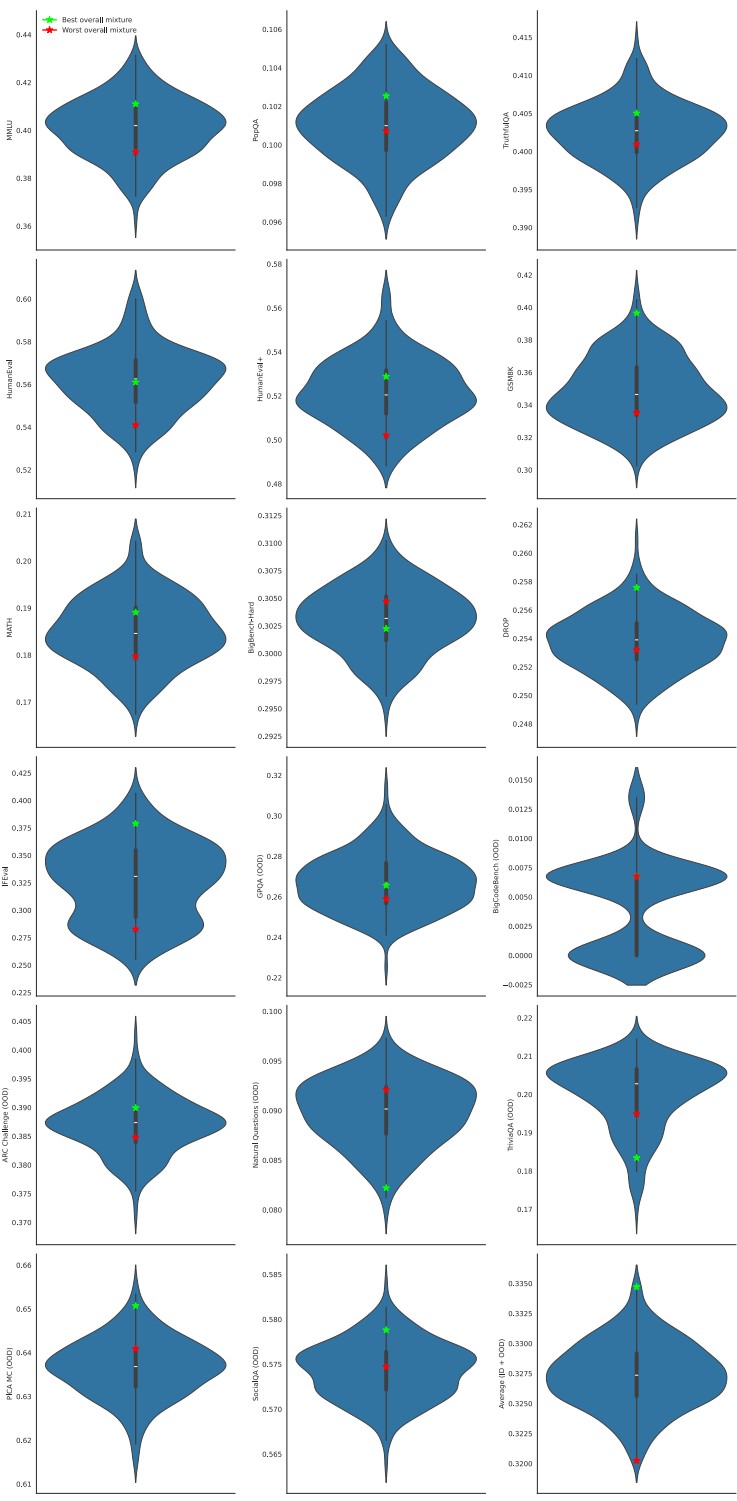

Figure 9: Violin plots of 0.5B models

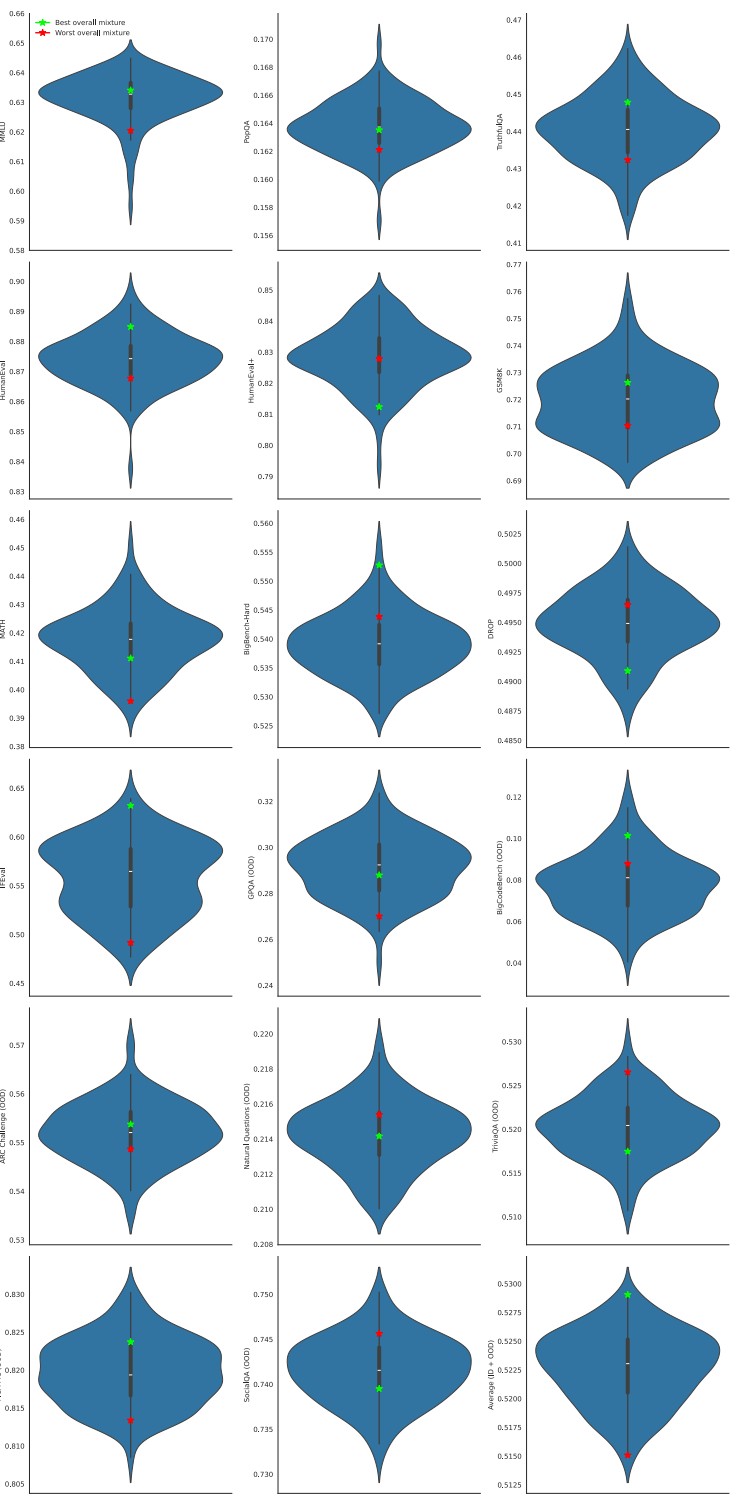

Figure 10: Violin plots of 3B models

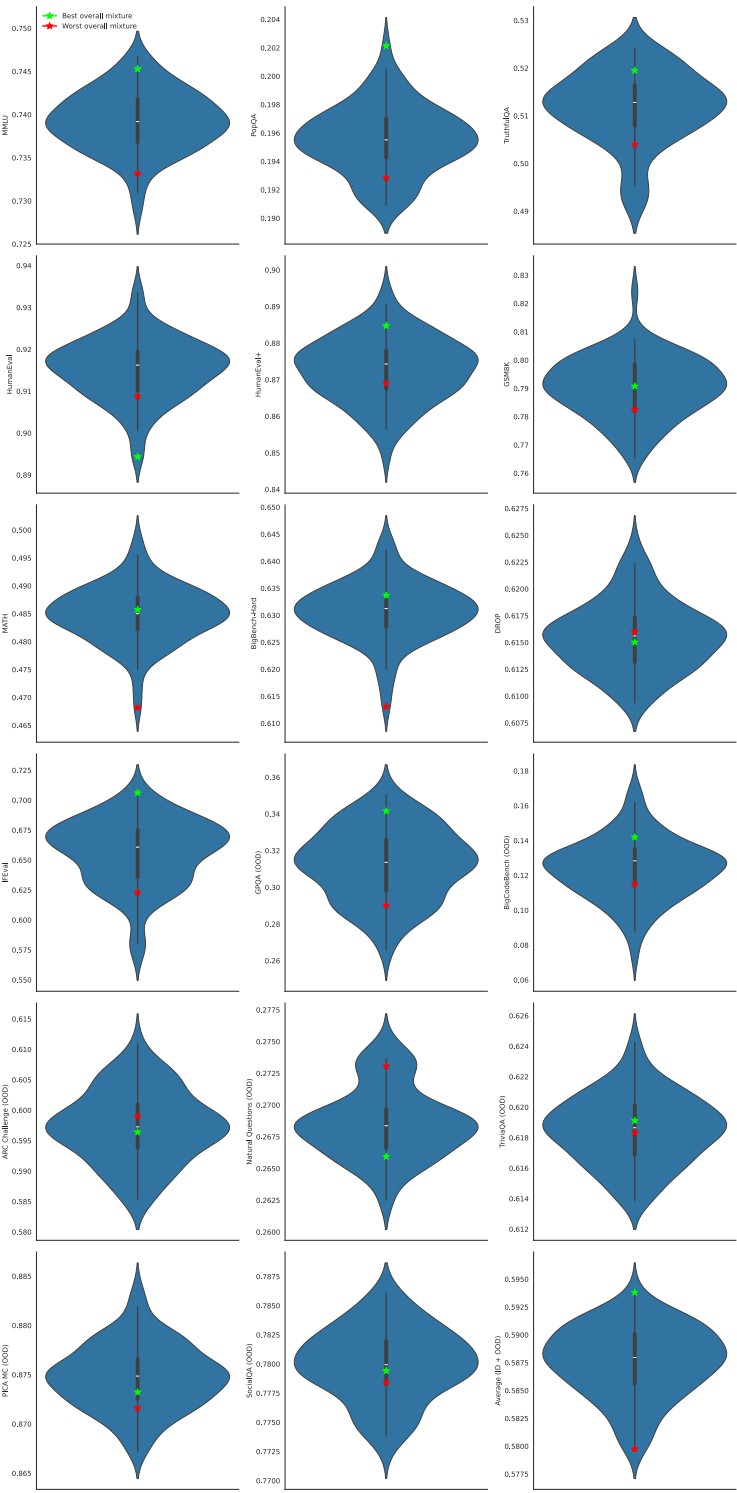

Figure 11: Violin plots of 7B models

