# OpenReview forum: "ADMIRE-BayesOpt: Accelerated Data MIxture RE-weighting for Language Models with Bayesian Optimization"
_TMLR — Accepted by TMLR_

### Review · Reviewer_i1bg · 2025-09-28

**Summary Of Contributions:**

This paper presents ADMIRE-BayesOpt, a Bayesian optimization framework for data mixture reweighting in language model training. It formulates validation performance under different data mixtures as a black-box function and models it using a multi-fidelity Gaussian Process (MF-GP) across multiple model scales. The framework incorporates structure-aware modeling to capture correlations among downstream tasks and employs a cost-sensitive acquisition strategy based on Max-value Entropy Search (MES) to efficiently explore the mixture space. Experiments on both pretraining (The Pile) and instruction fine-tuning (Tülu 3) demonstrate that ADMIRE-BayesOpt identifies effective mixtures with fewer training runs than baselines such as random search, RegMix, DoReMi, and fixed heuristics.

Strengths
- The paper addresses the problem of determining effective data mixture strategies for language model training by formulating it as a black-box optimization problem, where mixture proportions are optimized based on validation performance.
- It presents a Bayesian optimization framework that incorporates model outputs from different parameter scales (e.g., 125M to 6.9B) via a multi-fidelity Gaussian Process, with the goal of improving sample efficiency under limited training budgets.
- The framework includes a task-aware modeling component intended to capture correlations between validation objectives.
- Empirical results on both pretraining and fine-tuning tasks indicate that the method often achieves better performance and efficiency compared to heuristic and learning-based baselines.

Weaknesses
- The acquisition function (Max-value Entropy Search) is introduced without an empirical comparison to standard alternatives (e.g., UCB, EI), making it unclear how critical this design choice is to the method’s performance.
- The method assumes a fixed architecture family and validation setup, which may limit its applicability in scenarios involving continual learning, evolving data sources, or architectural heterogeneity.
- No theoretical analysis is provided on the optimization behavior, such as convergence rate, regret bounds, or sample complexity, which may be relevant for assessing robustness across settings.

**Audience:**

Yes

**Audience Explanation:**

The paper addresses a practical and increasingly relevant challenge in large-scale language model training—how to optimize data mixture weights effectively. By casting this as a black-box optimization problem and leveraging multi-fidelity modeling across model sizes, the proposed approach provides a systematic and resource-aware framework that differs from existing heuristic or hand-tuned strategies. These findings would likely interest researchers working on data-centric optimization, efficient training pipelines, and AutoML for foundation models.

**Claims And Evidence:**

Yes

**Claims Explanation:**

The main claims are supported by consistent results across pretraining and instruction tuning tasks, showing improvements over baselines like RegMix and DoReMi. Multi-fidelity modeling is empirically validated, but other components—such as task structure and acquisition function—lack isolated evaluation. While wall-clock cost is reported, the internal computational overhead from Bayesian Optimization is not analyzed. Overall, the evidence is credible but could benefit from deeper ablations and cost breakdowns.

**Requested Changes:**

- Please consider adding empirical comparisons for the acquisition function (e.g., UCB, EI), as noted in the weaknesses section.
- Provide an analysis of the internal computational cost of BO components, such as GP inference and acquisition evaluation.

---

> ### Author Response · Authors · 2025-10-15
> **Response to Reviewer i1bg**
>
> We sincerely thank Reviewer i1bg for their thorough and constructive review of our work. We are delighted to hear your positive feedback about our systematic approach to data mixture optimization and the consistent empirical improvements demonstrated across both pretraining and instruction tuning tasks.
>
> We have carefully considered all the weaknesses and requested changes identified in the review, and we address each point-by-point in our response below. We will revise the manuscript accordingly and incorporate these changes following your suggestions, which we believe will strengthen our paper and clarify the design choices underlying our method. We are happy to engage in further discussion and provide any additional clarifications if there are further questions.

---

> ### Author Response · Authors · 2025-10-15
> **[W1 + C1] Acquisition Function**
>
> >The acquisition function (Max-value Entropy Search) is introduced without an empirical comparison to standard alternatives (e.g., UCB, EI), making it unclear how critical this design choice is to the method’s performance.
>
> We would like to first highlight our motivation for the acquisition function choices in our proposed method. Our choice of Max-value Entropy Search (MES) stems from **established best practices in the MFBO literature [1,2]**, where MES-based acquisition functions have demonstrated superior empirical performance in multi-fidelity settings compared to traditional alternatives. This is because MES **directly optimizes information gain** about the maximum value across fidelities, which is particularly well-suited for our setting where we query models at different scales. **More crucially, this choice is also justified through our empirical ablation study below**.
>
> >Please consider adding empirical comparisons for the acquisition function (e.g., UCB, EI), as noted in the weaknesses section.
>
>
> We sincerely appreciate this valuable suggestion, which has helped strengthen our empirical validation. Following the reviewer's recommendation, we have conducted an additional **ablation study comparing different acquisition function choices**:
>
> **Experimental setup:** We evaluate acquisition functions in two settings:
> - For **ADMIRE-BayesOpt**, we compare our chosen Expected Improvement (EI) against standard alternatives including Upper Confidence Bound (UCB) and Probability of Improvement (PI) in the no-transfer setting, which directly optimizes data mixtures on the 7B target model using our ADMIRE-IFT dataset.
> - For **ADMIRE-MFBO**, we compare MF-MES against multi-fidelity variants (CostAwarePI, CostAwareUCB).
> - In both settings, we report results up to 30,000 minutes of wall-clock time. For reference, the optimal data mixture in the ADMIRE-IFT dataset achieves an evaluation score of 0.594, which serves as the performance upper bound for these experiments.
>
> | ADMIRE-BayesOpt Acquisition Function | Best Performance ($\uparrow$) | Time to Best (min) ($\downarrow$) |
> |---------------------|-----------------|-------------|
> | EI (Ours)        | **0.593**       | 5912  |
> | PI             | 0.591           | **537**  |
> | UCB             | 0.590           | 537      |
>
> | ADMIRE-MFBO Acquisition Function | Best Performance ($\uparrow$) | Time to Best (min) ($\downarrow$) |
> |---------------------|-----------------|-------------|
> | MF-MES (Ours)          | **0.594**       | 5651   |
> | CostAwarePI              | 0.594        | 26293      |
> | CostAwareUCB               | 0.592           | **953**       |
>
> **Key observations from the ablation study:**
> - **MF-MES achieves the best final performance** (0.594), successfully identifying the optimal data mixture, while other acquisition functions plateau at suboptimal solutions (0.592).
> - The multi-fidelity variants (CostAwarePI, CostAwareUCB) perform better than their single-fidelity counterparts, confirming the value of leveraging multiple model scales.
> - However, **MF-MES's information-theoretic approach to balancing exploration and exploitation across fidelities** proves most effective for our data mixture optimization problem, justifying our design choice.
>
> We will include this ablation study in our revised manuscript.
>
> #### References
> [1] Takeno et al., Multi-fidelity Bayesian Optimization with Max-value Entropy Search, ICML, 2020
>
> [2] Moss et al., GIBBON: General-purpose Information-Based Bayesian Optimisation, JMLR, 2021

---

> ### Author Response · Authors · 2025-10-15
> **[W2] Assumptions on Fixed Architecture and Validation Setup**
>
> ### Multi-fidelity modeling and assumption on architectural similarity:
> >The method assumes a fixed architecture family and validation setup,...
>
> We thank the reviewer for this thoughtful observation regarding the assumptions underlying our framework. We would like to respectfully clarify that MFBO fundamentally relies on **correlation between different fidelities** [1,2], where lower-fidelity evaluations provide informative approximations of the expensive target objective. In our work, this correlation is naturally achieved through models of different scales within the same architecture family. As **empirically demonstrated in Section 6.1.3**, these models exhibit sufficient performance correlation to enable effective transferability, allowing the multi-fidelity approach to successfully guide optimization toward the target objective.
>
> This setup is also **consistent with existing data mixture optimization methods** such as DoReMi[3] and DataMixingLaw[4], which similarly rely on smaller proxy models from the same family to guide mixture selection for larger target models. The empirical success of these approaches—and our own results —demonstrates that this assumption is both **reasonable and effective in practice**.
>
> ---
>
> ### Applicability to dynamic environments:
> > ..., which may limit its applicability in scenarios involving continual learning, evolving data sources, or architectural heterogeneity
>
> Regarding scenarios involving continual learning, evolving data sources, or architectural heterogeneity, we fully acknowledge these as important limitations that affect not only ADMIRE-BayesOpt but also all baseline approaches in the current literature.
>
> We greatly appreciate the reviewer for highlighting this gap, as it points to a **valuable direction for future research**.
>
> In response to this feedback, we will **expand the Discussion in our revised manuscript** to explicitly address these limitations and outline potential approaches to handle dynamic environments. We include **some of our key points** discussed below:
>
> - **Adapting our method to evolving training data sources**: When a new domain becomes available during training, ADMIRE-BayesOpt **does not need to restart from scratch**. Instead, the optimization can continue by **expanding the mixture weight dimensionality** to include the new domain. **All previously evaluated configurations remain valid observations** in the expanded space, with the new domain weight set to zero. The GP surrogate model will initially have **high uncertainty** around the performance of mixtures involving the new domain, as predictions represent extrapolation from observed boundary cases. Consequently, the acquisition function will guide targeted exploration of the expanded space, requiring **fewer additional optimization steps** than complete re-optimization while building confidence in regions where the new domain contributes meaningfully.
>
> - **Addressing more sophisticated dynamic environments**: Beyond the mentioned approach above, we discuss potential solutions including:
>   - **Meta-learning or transfer learning approaches** to initialize the Gaussian Process with informative priors when training or evaluation distributions shift, enabling faster adaptation to new environments [5]
>   - **Online Bayesian optimization** strategies that can incrementally update the GP surrogate model in a time-varying environment [6]
>   - **More sophisticated kernel designs** for the GP to handle and generalize to architectural heterogeneity across model families [7]
>
> We believe that **explicitly documenting these limitations and future directions** will not only strengthen the paper but also provide the community with concrete pathways for extending this line of work. We hope this addition addresses the reviewer's concern while contributing to a more complete understanding of the method's scope and potential.
>
> #### References
> [1] Takeno et al., Multi-fidelity Bayesian Optimization with Max-value Entropy Search, ICML, 2020\
> [2] Moss et al., GIBBON: General-purpose Information-Based Bayesian Optimisation, JMLR, 2021\
> [3] Xie et al., DoReMi: Optimizing Data Mixtures Speeds Up Language Model Pretraining, NeurIPS, 2023\
> [4] Ye et al., Data Mixing Laws: Optimizing Data Mixtures by Predicting Language Modeling Performance, arXiv, 2024\
> [5] Wei et al., Meta-learning Hyperparameter Performance Prediction with Neural Processes, ICML 2021\
> [6] Bogunovic et al., Time-Varying Gaussian Process Bandit Optimization, AISTATS, 2016\
> [7] Wilson et al., Deep Kernel Learning, AISTATS, 2016

---

> ### Author Response · Authors · 2025-10-15
> **[W3] Theoretical Analysis**
>
> >No theoretical analysis is provided on the optimization behavior, such as convergence rate, regret bounds, or sample complexity, which may be relevant for assessing robustness across settings
>
>
> We appreciate the reviewer raising this important consideration. We **fully agree** that deriving novel regret bounds for multi-fidelity MES would be valuable. However, we respectfully note that deriving such guarantees for multi-fidelity MES represents, to the best of knowledge, an **ongoing area of active research** within the BO community and is **beyond the scope** of this work.
>
>
> Our primary **scope and contributions** are to **demonstrate that Bayesian optimization methods are well-suited** to the problem of data mixture optimization in language model training, and to **develop a concrete, practical implementation (ADMIRE-BayesOpt and ADMIRE-MFBO)** based on MFBO techniques that achieves significant speed-ups in finding optimal data-mixtures for LLM training compared to previous baselines.
>
> Regarding theoretical guarantess for ADMIRE-BayesOpt and ADMIRE-MFBO:
> - **For standard BO components**: When using ADMIRE-BayesOpt with classical acquisition functions like Expected Improvement, well-established regret bounds and convergence guarantees directly translate to ADMIRE-BayesOpt.
> - **For ADMIRE-MFBO with multi-fidelity Max-value Entropy Search (MF-MES)**: Deriving comprehensive regret bounds for MF-MES is, to the best of our knowledge, an **ongoing area of active research** within the BO community. While foundational work exists, complete theoretical results remains an open problem:
>   -  Wang & Jegelka [1] establish regret bounds for single-fidelity MES with specific sampling constraints (Monte Carlo integration sample size of 1)
>   -  Zhang et al. [2] consider multi-fidelity and multi-task BO and extend the MES result to the single-fidelity multi-task setting
>   -  Song et al. [3] provide regret bounds for an episodic two-stage multi-fidelity algorithm where each episode separately collects low-fidelity data (stage 1) then high-fidelity data (stage 2), which differs from the continuous multi-fidelity setting we employ
>   -  There also exist multiple implementations of MES (GIBBON [4] and VES [5]) that use different mathematical approximations for which convergence guarantees are still an open question.
>
>
>
> We will add the discussion above in our revised manuscript acknowledging the current state of theoretical guarantees for MF-MES and MFBO and **highlighting that extending these guarantees represents an important direction for future work**, both for the BO community and for strengthening the theoretical foundations of data mixture optimization methods.
>
> #### References:
> [1] Wang & Jegelka, Max Value Entropy Search For Efficient Bayesian Optimization, ICML 2017\
> [2] Zhang et. al., Multi-Fidelity Bayesian Optimization with Across Task Transferable Max-Value Entropy Search, IEEE transactions on Signal Processing 2025\
> [3] Song et.al., A General Framework for Multi-Fidelity Bayeisan Optimization with Gaussian Processes, AISTATS 2019\
> [4] Moss et.al., GIBBON: General-purpose Information-Based Bayesian OptimisatioN, JMLR 2021\
> [5] Cheng et.al., A Unified Framework for Entropy Search and Expected Improvement in Bayesian Optimization, ICML 2025

---

### Review · Reviewer_esBp · 2025-09-28

**Summary Of Contributions:**

This paper proposes an interesting approach to optimize data mixture to train LLMs using Bayesian Optimization.

Given M models at different complexity scales (including the target LLM with high complexity) and d datasets, the proposed method  iteratively schedules information-acquisition experiments (pi, m) where m is a model in the model set and pi is a (discrete) mixture distribution over d data sources. Each experiment thus allows the method to acquire an (experiment, outcome) data point where experiment corresponds to the tuple (pi, m) and outcome corresponds to the empirical performance of model m after it has been updated via further training with the re-weighted data mixture pi. This data point will be used to update a Gaussian process model that maps from experiment to outcome.

Leveraging this Gaussian process at each iteration, one can use standard BO acquisition functions such as Expected Improvement (EI) and Maximum Entropy Search (MES) to schedule the next experiment (pi, m). Once the experiment budget has been used up, the mixture pi that maximizes the GP predictive mean at (pi, m = target LLM) will be selected as the optimal data mixture.

The paper has run extensive experiments to validate this approach based on 460 LLMs from a Qwen2.5 model family with sizes (500m, 3b, 7b) on 256 diverse data mixtures, with 13K+ GPU hours on A100. The whole set of experimental run across BO iterations has also been released as open dataset. The experiments also include OOD, zero-shot transfer, and multi-fidelity evaluations

**Audience:**

Yes

**Audience Explanation:**

Effective training of LLM requires a lot of delicate calibration and optimizing for data mixture is one such important issue. This paper has formulated an elegant solution for it using classical methods in Bayesian Optimization. I believe this is valuable and will be of interest to both the BO and LLM community.

**Claims And Evidence:**

Yes

**Claims Explanation:**

Overall, I believe this is a valuable contribution even though it does not introduce any new algorithms. The impact here, in my opinion, is of practical significance. A very practical problem of optimizing data mixture for LLM training has been nicely formulated from the lens of Bayesian optimization which has been carefully validated through extensive experiment validation. I really like the extensive experimentation filled with rich insights. Another plus is that the authors have released all model training, fine-tuning, and evaluations across the BO runs as open dataset which could really benefit the community.

My relatively minor concerns with this paper are:

First, can the authors please discuss the advantage of the method over reinforcement learning (RL) approach? Alternative to the BO approach, RL methods (e.g., actor-critic) can also be used to optimize this data mixture. Will it be more/less effective in terms of performance and/or compute/storage cost? From a practical point of view, updating the GP is costly and there is also a risk of kernel misspecification (see below).

Second, can the authors please explain why such simplified kernel choices (on no. of parameters) would suffice? Maybe it fits for the specific settings here but it likely won't in more sophisticated cases. For example, I could create a deep model with just feed-forward units to match the complexity of any existing LLMs in the model set. The current kernel choice would determine that the LLM and the artificially created model are highly correlated in performance but it is likely not true as the pure feed-forward model will not be able to match the transformer-based LLM's performance.

Third, given the acquisition function, how exactly do the authors solve for the mixture? This seems to lead to a constrained optimization task and it seems the manuscript glosses over such details. Including these details in the appendix would be helpful.

**Requested Changes:**

My requested changes follow the weaknesses I highlighted above:

1. Positioning/comparing with existing RL approaches. Highlight more clearly the advantage of BO over RL in this specific scenarios

2. Discuss the choice of the model kernel.

3. Including details regarding the optimization of the acquisition function.

---

> ### Author Response · Authors · 2025-10-15
> **Response to Reviewer esBp**
>
> We sincerely thank Reviewer esBp for the thorough and thoughtful review of our work. We greatly appreciate your recognition of its practical significance, the extensive experimental validation, and the value of our open dataset release to the research community. Your positive assessment is very encouraging.
>
> We have carefully considered each of your concerns and provide detailed responses to the three weaknesses and requested changes in the comments below. We will include all discussion and analysis from our rebuttal responses in the final revised manuscript. We welcome any further questions and are happy to continue the discussion.

---

> ### Author Response · Authors · 2025-10-15
> **[W1+C1] Bayesian Optimization vs Reinforcement Learning**
>
> >First, can the authors please discuss the advantage of the method over reinforcement learning (RL) approach? [....] Will it be more/less effective in terms of performance and/or compute/storage cost? [...] Positioning/comparing with existing RL approaches. Highlight more clearly the advantage of BO over RL in this specific scenarios
>
> Thank you for this valuable question. We **fully agree** that in theory, RL approaches could be applied to data mixture optimization as a black-box optimization problem. However, we would like to emphasize that **BO is more data-efficient and thus the preferable choice for the practical settings we target** in this work.
>
> The critical constraint in data mixture optimization is the **extreme low-data regime**.
> - Each observation requires training a language model to convergence on a specific data mixture and evaluating it on downstream benchmarks—an operation that **demands significant computational resources**, consuming hours to days of GPU time.
> - In practice, **practitioners can only afford tens to perhaps a hundred such observations** before the optimization cost becomes prohibitive. This constraint fundamentally shapes which optimization methods are viable.
>
> **RL approaches are poorly suited to this regime**:
> - Methods like actor-critic require **substantial sample complexity** to learn effective policies, typically needing hundreds to thousands of interactions to simultaneously learn both the policy (actor) and value function (critic).
> - With only a handful of observations available, **RL methods could exhaust most of their budget in early exploration** without converging to strong solutions.
> - Furthermore, RL policies trained on such limited data **tend to overfit and fail to generalize**.
>
> **Bayesian optimization, by contrast, is well-suited for this setting**:
> - The **GP surrogate model explicitly quantifies uncertainty** over the unknown objective landscape and uses this uncertainty to **guide efficient and effective exploration** through principled acquisition functions.
> - This allows BO to maximize information gain from each precious evaluation, learning the complex relationship between data mixture weights and model performance within a reasonable compute budget.
> - Where RL relies on trial-and-error learning that requires abundant data, BO's **probabilistic modeling enables principled decision-making** even with limited observations.
>
> **The superior sample efficiency of BO over RL is therefore the determining factor** for data mixture optimization, making BO the more practical and preferable choice here for this problem. We will include the discussion and comparison here in our revised manuscript.
>
> ---
>
> ### Cost breakdown of updating GPs
>
> >From a practical point of view, updating the GP is costly and there is also a risk of kernel misspecification (see below).
>
> We appreciate the opportunity to clarify the computational efficiency of our framework. Given that data-mixture optimization is **in the low-data regime, updating the GP is quite efficient**.
>
> During the rebuttal period, we conducted a detailed analysis of the internal computational cost breakdown for our method. We will include this analysis in our revised manuscript. We would like to highlight the following **key observations**:
>
> 1. **LLM training dominates** the total computational budget, while all Bayesian optimization components related to **our method incur only negligible overhead at less than <0.01%** per optimization iteration. Importantly, LLM training is unavoidable across all baseline data mixture optimization methods.
> 2. This breakdown reinforces the **core motivation of our work**: since LLM training is the overwhelming bottleneck in data mixture optimization, even modest reductions in the number of required training runs translate to substantial wall-clock time savings. Our method achieves exactly this—**minimizing the number of expensive LLM training iterations needed to identify optimal mixtures**, making the minimal BO overhead highly worthwhile.
> 3. Our empirical results demonstrate that ADMIRE-BayesOpt achieves **superior compute-performance tradeoffs and significant speedups (hence cost reduction)** compared to all baselines, as shown in our main results, confirming the **efficiency and practical feasibility** of our approach.
>
> We measured the **average wall-clock time** for each Bayesian optimization sub-step and calculated the **percentage cost** it occupies within one complete data mixture optimization iteration. Results are averaged over 100 optimization iterations:
>
>
> | **BO Compoent** | **Wall-Clock Time** | **Percentage Cost** |
> |------------------|---------------------|----------------------|
> | GP Posterior Update |  2.95 sec | <0.01% |
> | Acquisition Optimization |  0.043 sec | <0.01% |
> | **Non-BO Component** |  **Wall-Clock Time** | **Percentage Cost** |
> | LLM Training & Evaluation |  ~12 hour for 7B model | 99.9% |

---

> ### Author Response · Authors · 2025-10-15
> **[W2+C2] Model Kernel Choices (part 1/2)**
>
> Thank you for this insightful question regarding our kernel design. We appreciate the opportunity to provide further clarification and empirical validation.
>
> ### Regarding the role of our model kernel in MFBO
> >Second, can the authors please explain why such simplified kernel choices (on no. of parameters) would suffice?
>
> Our model kernel is designed to measure the **correlation in performance trends between models**, rather than their absolute performance levels. Specifically, our parameter-based kernel (downsampling kernel, Equation 3) **encodes a prior assumption** that **models with similar sizes**, when trained on the same data mixture, will **generally exhibit correlated performance trends**—meaning that when one model's performance increases with a particular mixture, another model of similar size is likely to show a similar directional improvement, **even though their absolute performance values may differ substantially**.
>
> This prior assumption is **grounded in established literature** demonstrating the relationship between model scale and performance correlation **[1,2,3]**. More importantly, this assumption is **empirically validated** throughout our experiments:
>
> - In **Section 6.1.3** (Increased Transferability with Proxy Model Scale), ADMIRE-BayesOpt progressively achieves **better transferability as proxy model size approaches the target model size**: reaching final performances of 59.19→59.35 for 0.5B→7B proxy models. This demonstrates that models closer in size indeed exhibit higher correlation in behavior, leading to improved transferability of the optimized mixture.
> - **ADMIRE-MFBO consistently achieves optimal compute-performance trade-offs** across our extensive experiments, providing strong evidence for the practical effectiveness of our kernel choice.
>
> >Maybe it fits for the specific settings here but it likely won't in more sophisticated cases.
>
> We would like to respectfully clarify that our MFBO framework is designed for **the most realistic and practical settings in data mixture optimization-specifically**, scenarios where:
> - Practitioners use architecturally similar models (e.g., smaller models from the same family to predict larger model performance), which is a well-established approach in practical applications [2,3].
> - Practical data-mixture optimization operates in an **extreme low-data regime** where the GP regression problem has only a handful of training data points **due to the prohibitive cost of training LLMs**. For example, collecting the entire 460 model training data points in our ADMIRE-IFT benchmark requires a total of 13,000 A100 GPU hours!
>
> In such realistic, low-data regimes, our **simpler kernel with strong inductive bias is not only data-efficient but preferable—it generalizes better compared to sophisticated alternatives** (e.g., neural network kernels) that would require substantially more observations to be properly trained and carry high risk of overfitting.
>
>
> ### Regarding the feed-forward vs. Transformer example
> >For example, I could create a deep model with just feed-forward units to match the complexity of any existing LLMs in the model set. The current kernel choice would determine that the LLM and the artificially created model are highly correlated in performance but it is likely not true as the pure feed-forward model will not be able to match the transformer-based LLM's performance.
> >
>
> Thank you for raising this interesting point about extreme architectural mismatches. While such scenarios might fall outside the practical use cases our framework targets (as discussed above), we would like to clarify **how our Bayesian framework handles even such extreme cases**:
> - While our model kernel in Equation 3 encodes an **initial prior assumption** based on parameter count, this prior is **continuously refined through Bayesian posterior updates** (Equations 4-5) as we gather realistic observations during online data mixture optimization.
> - In cases of extreme architectural mismatch (such as your feed-forward example), with more obeservations, **the GP posterior can update** (learnable kernel parameters $c$ and $\delta$) **to incorporate the empirical evidence that specific proxy models behave differently than the prior kernel suggests**, with the posterior mean predictions increasingly weighted toward the empirical observations rather than the prior assumptions.

---

> ### Author Response · Authors · 2025-10-15
> **[W2+C2] Model Kernel Choices (part 2/2)**
>
> ### Additional ablation study on model kernel choices for MFBO
> To further strengthen our kernel design justification for MFBO, we conducted additional ablation studies during the rebuttal period, comparing our downsampling model kernel against alternative formulations while keeping everything else identical:
>
>
> | ADMIRE-MFBO Kernel | Best Performance ($\uparrow$) | Time to Best (min) ($\downarrow$) |
> |---|---|---|
> | RBF+downsampling (Ours) | **0.594** | **5651** |
> | RBF+RBF | 0.594 | 17500 |
> | Matern+downsampling | 0.592 | 6365 |
>
> These results demonstrate that:
> - Switching the model kernel from **downsampling $\rightarrow$ RBF** results in **worse performance** and significantly **higher computational cost**.
> - our **RBF+downsampling** kernel consistently achieves the **the best performance with minimal computational cost** in wall-clock time.
>
> We greatly appreciate the reviewer's comment, and **we acknowledge that exploring more sophisticated kernel designs** (e.g., incorporating architectural features such as depth, attention mechanisms, or training dynamics) **is indeed an important direction for future work**. The ablation study and extended discussion on kernel design considerations will be added to the revised manuscript.
>
> #### References
> [1] Kaplan et al., Scaling Laws for Neural Language Models, arXiv, 2020\
> [2] Xie et al., DoReMi: Optimizing Data Mixtures Speeds Up Language Model Pretraining, NeurIPS, 2023\
> [3] Ye et al., Data Mixing Laws: Optimizing Data Mixtures by Predicting Language Modeling Performance, arXiv, 2024

---

> ### Author Response · Authors · 2025-10-15
> **[W3+C3] Acquisition Function Optimization Details**
>
> > Third, given the acquisition function, how exactly do the authors solve for the mixture? This seems to lead to a constrained optimization task and it seems the manuscript glosses over such details. Including these details in the appendix would be helpful.
>
> We thank the reviewer for this valuable suggestion. We will include the following detailed explanations in the Appendix of our revised manuscript to address this important implementation detail.
>
> ### Find the next data mixture during acquisition:
>
> As the reviewer correctly notes, given an acquisition function $\alpha_t(\pi, m)$ (e.g., Expected Improvement or Maximum Entropy Search as described in Section 3.2), finding the next query point $(\pi\_{t+1}^\*, m\_{t+1}^\*)$ requires solving a **constrained optimization problem** in practice:
>
> $$(\pi_{t+1}^\*, m_{t+1}^\*) = \arg\max_{\pi, m} \alpha_t(\pi, m)$$
>
> subject to the **simplex constraint** that $\pi \in \Delta^{d}$ (i.e., $\sum_{i=1}^{d} \pi_i = 1, \pi_i \geq 0$ for all $i \in \{1,\ldots,d\}$), ensuring that the mixture proportions sum to 1.
>
> While this constrained optimization can be handled by various off-the-shelf methods [2,3] (which are orthogonal to the main focus of our paper), **we describe one specific implementation below** for completeness:
> 1. **Enumeration over model candidates**: Since $m$ is discrete and finite (i.e., $|\mathcal{M}| = 3$ models for multi-fidelity BO, or $|\mathcal{M}| = 1$ for standard BO), we enumerate over all candidate proxy models. For each proxy model $m \in \mathcal{M}$, we perform the mixture optimization described in step 2 below.
> 2. **Mixture optimization via projected gradient ascent**: For the continuous variable $\pi$ constrained to the probability simplex, we optimize $\alpha_t(\pi, m)$ given $m$ using **projected gradient ascent with the simplex projection operator**. This iterative procedure consists of the following sub-steps:
>    - **Unconstrained Gradient ascent step**: At iteration $j$, compute: $\tilde{\pi}^{(j+1)} = \pi^{(j)} + \eta \nabla_{\pi} \alpha_t(\pi^{(j)}, m)$, where $\eta$ is the learning rate.
>    - **Projection step**: Project $\tilde{\pi}^{(j+1)}$ back onto the probability simplex to obtain the next iterate: $\pi^{(t+1)} = \text{Proj}_{\Delta^{d}}(\tilde{\pi}^{(t+1)})$. This can be implemented using the efficient Euclidean projection described in Duchi et al. [1], or alternatively via the softmax operation.
>
>    The projection ensures that iterates remain feasible while moving in the direction of steepest ascent.
>
> 3. **Selection of optimal query point**: After optimizing $\alpha_t(\pi, m)$ over $\pi$ for each model $m \in \mathcal{M}$, we select the $(\pi^\*\_{t+1}, m^\*\_{t+1})$ pair that achieves the **maximum acquisition value** across all model candidates:
>    $$(\pi^\*\_{t+1}, m^\*\_{t+1}) = \arg\max_{m \in \mathcal{M}} \max_{\pi \in \Delta^{d}} \alpha_t(\pi, m)$$
>
> As shown in **Figure 1 and Section 4.1**, we then train model $m\_{t+1}^\*$ with mixture $\pi_{t+1}^\*$, evaluate its performance to obtain a new observation, update the GP posterior mean and variance following **Equations 4-5**, and proceed to the next iteration.
>
> ### Recommend a final optimal data mixture:
>
> At any iteration $t$,  we can recommend a single optimal data mixture for the target model $m'$ by **optimizing the fitted Bayesian posterior mean** $\mu_t(\pi, m')$:
>
> $$\pi^* = \arg\max_{\pi \in \Delta^{d}} \mu_t(\pi, m')$$
>
> This optimization employs the **same projected gradient ascent procedure** with simplex projection described above, following standard practice in Bayesian optimization literature [2,3]. Importantly, **this exploits our accumulated posterior knowledge about data mixture performance** rather than the acquisition function.
>
>
> ### Experimental implementation:
> We note that for our experimental evaluation in Section 5, we evaluate our method **on the same discrete set of mixture candidates used in established datasets** to enable direct and fair comparison with prior work [4]. Specifically, we evaluate the acquisition function $\alpha_t(\pi, m)$ only at the candidate mixtures included in the benchmark. We then select the data mixture that yields the **highest acquisition value** parameterized by our GP posterior for the next query, or the **highest GP posterior mean** $\mu_t(\pi, m')$ for the final recommendation. The continuous optimization framework described above represents the general formulation of our method and can be readily applied when working outside the constraints of benchmark datasets with discrete mixture candidates.
>
> #### References
> [1] Duchi et al., Efficient Projections onto the ℓ1-Ball for Learning in High Dimensions, ICML, 2008\
> [2] Frazier, A Tutorial on Bayesian Optimization, arXiv, 2018\
> [3] Garnett, Bayesian Optimization, Cambridge University Press, 2023\
> [4] Liu et al., RegMix: Data Mixture as Regression for Language Model Pre-training, ICLR, 2025

---

### Review · Reviewer_5EU5 · 2025-10-04

**Summary Of Contributions:**

This paper studies data mixture problem, the goal of which is finding a mixing parameter for datasets that yields best final performance of large models. The approach is to view the data mixture problem as a regression problem of mixing parameter $\pi$ and model $m$, and the authors solve this problem using Bayesian optimization. The proposed method, namely ADMIRE-BayesOpt, solves Gaussian process for estimating validation score with RBF and Downsampling (only for multi-fidelity setting) kernels. The core assumption behind applying the Gaussian process is smoothness of the relationship between the pair of mixing parameter in simplex and proxy model and the validation score of large model. Throughout the experiments with strong baseline methods, the authors validate the superiority of the proposed method in terms of both cost efficiency and performance. Furthermore, the authors release the benchmark dataset for broadening the research in this field.

**Audience:**

Yes

**Audience Explanation:**

The problem is directly related to ML society, especially optimizing LLM.

**Claims And Evidence:**

Yes

**Claims Explanation:**

The paper is well-written, organized and easy to read. The problem formulation and applying the Bayesian optimization are reasonable and correct, and experimental results validate the proposed approach. In particular, they beat all the baseline methods in their setting, which would be interesting to the community working on this problem.

**Requested Changes:**

While I have enjoyed reading the paper, I have some questions and comments:
1. Could you add the validity of smoothness between the parameter $(\pi, m)$ and the validation score $y$ from your experiments? Since applying Gaussian process is grounded by smoothness assumption, it would be more convincing if any experimental results verify this assumption.
2. Why MFMS-GP (Yen et al., 2025) is compared only in Figure 7? Adding the results from MFMS-GP on Figure 5 and 6 would make the paper more strong and convincing.
3. The paper explains more about MFMS-GP (Yen et al., 2025) with details (in terms of methodology) as the method uses similar approaches.

---

> ### Author Response · Authors · 2025-10-15
> **Response to Reviewer 5EU5**
>
> We sincerely thank Reviewer 5EU5 for the thoughtful and constructive feedback. We are delighted that you found our paper enjoyable to read and appreciate the time and effort you invested in providing detailed suggestions for improvement.
>
> We have carefully considered each of your comments and believe they have helped strengthen the manuscript significantly. Below, we provide detailed responses to each point you raised, organized by the weakness/question numbers from your review. All suggested changes will be incorporated into the revised manuscript. Please let us know whether our responses adequately address your concerns.

---

> ### Author Response · Authors · 2025-10-15
> **[C1] Our Assumption in GPs**
>
> >Could you add the validity of smoothness between the parameter and the validation score from your experiments? Since applying Gaussian process is grounded by smoothness assumption, it would be more convincing if any experimental results verify this assumption.
>
> We thank the reviewer for this insightful question and appreciate the opportunity to clarify the theoretical requirements and provide empirical validation.
>
> ---
>
> ### Correlation structure vs. strict smoothness
> We would like to clarify the fundamental assumptions underlying GPs in our work: While smoothness is often associated with GPs, it is not a strict mathematical requirement. More precisely, GPs rely on **exploitable correlation structure** in the objective function, which is **encoded through the choice of kernel function**. The essential assumption is that nearby points in the input space tend to have correlated output values—a form of local structure that the GP can exploit for prediction. The kernel function determines both the nature and degree of this correlation, and can accommodate functions with varying smoothness properties, **including non-smooth or locally irregular functions** [1].
>
> ---
>
> ### Empirical evidence
> Our experimental results provide **multiple independent lines of evidence** that the data mixture optimization problem exhibits the necessary correlation structure for effective GP modeling:
> 1. **Consistent Optimization Success Across Experiments**: Our Bayesian Optimization approach achieves systematic improvements over baselines and demonstrates over 500% speed-ups in finding optimal mixtures. If the objective function lacked sufficient correlation structure, the GP surrogate model would fail to make useful predictions.
> 2. **Zero-shot Transfer and MFBO Effectiveness (Figure 5&7)**: We observe effective knowledge transfer of mixture performance across model scales (from 1M to 7B parameters). This cross-scale transferability provides strong additional evidence of underlying correlation structure. Specifically, if the relationship between mixture weights and validation performance were highly irregular or lacked structure, we would not observe consistent patterns that generalize predictably across different model scales.
> 3. **Sensitivity Analysis (Figure 4)**: The sensitivity analysis presented in Figure 4 provides direct visual evidence of the correlation structure, where certain evaluation datasets exhibit noticeable sensitivity to mixture changes, showing interpretable relationships rather than random fluctuations and revealing the underlying correlation structure that is exploitable by GPs.
>
> ---
>
> ### Robustness and flexibility through kernel design
>
> We would also like to emphasize that **kernel choice provides significant flexibility** in accommodating different function properties. Our approach could accommodate more flexible kernels (e.g., spectral mixture kernels[2], neural network kernels[3]) for objective functions with different characteristics. This flexibility means that our approach does not critically depend on strong smoothness assumptions, but rather **leverages the more general and empirically validated principle that data mixture optimization exhibits exploitable correlation structure**.
>
> We sincerely appreciate the reviewer's comment, as it highlights an important clarification. We will revise our manuscript to explicitly mention our underlying assumption on exploitable correlation structure and include the empirical evidence presented above to validate that our problem domain exhibits the necessary properties for effective GP-based optimization.
>
> #### References:
> [1] Rasmussen, C. E., & Williams, C. K. I. Gaussian Processes for Machine Learning. MIT Press, 2006\
> [2] Wilson et al., Gaussian process kernels for pattern discovery and extrapolation, ICML 2013\
> [3] Wilson et al., Deep kernel learning, AISTATS 2016

---

> ### Author Response · Authors · 2025-10-15
> **[C2] Clarification on MFMS-GP in Figure 7**
>
> >Why MFMS-GP (Yen et al., 2025) is compared only in Figure 7? Adding the results from MFMS-GP on Figure 5 and 6 would make the paper more strong and convincing.
>
> Thank you for this suggestion. We would like to explain our rationale behind our experimental presentation.
>
> Figures 5 and 6 evaluate and compare methods in a zero-shot transfer setting with **a single proxy model**, while Figure 7 focuses on the **multi-fidelity** setting with multiple proxy models. These represent fundamentally different practical scenarios with different information resources available for data-mixture optimization. To maintain clarity, we deliberately separated these settings rather than mixing them within the same figures.
>
> **Regarding why MFMS-GP appears only in Figure 7**: MFMS-GP is specifically designed for multi-fidelity optimization and, unfortunately, the authors did not provide an off-the-shelf single-fidelity implementation that we could directly compare against in the standard Bayesian optimization setup of Figures 5-6.
>
> Instead, we include MFMS-GP in Figure 7, where we compare both ADMIRE-MFBO and MFMS-GP against **a unified, Pareto-optimal baseline** constructed from the element-wise best performance across all standard single-fidelity methods at each time step. This demonstrates both:
> - leveraging multiple proxy models (as both multi-fidelity methods do) provides genuine advantages over single-fidelity approaches
> - ADMIRE-MFBO's effectiveness relative to MFMS-GP
>
> We appreciate the reviewer pointing out that this organizational choice may not have been sufficiently clear. To improve readability and transparency, we will improve the figure captions to explicitly state the experimental setting for each comparison (zero-shot transfer with single proxy model in Figures 5-6, multi-fidelity setting with multiple proxy models in Figure 7).
>
> We hope this clarification addresses the reviewer's concern while maintaining the clarity and logical flow of our experimental presentation.

---

> ### Author Response · Authors · 2025-10-15
> **[C3] More Discussion about MFMS-GP**
>
> >The paper explains more about MFMS-GP (Yen et al., 2025) with details (in terms of methodology) as the method uses similar approaches.
>
> We appreciate the reviewer's request for a more discussion on comparison with MFMS-GP (Yen et al., 2025). We fully acknowledge MFMS-GP as concurrent work that independently explores Bayesian optimization for data mixture selection.
>
> While both works share this high-level motivation, **our method addresses the data-mixture optimization problem in a substantially more comprehensive and realistic setting**, with key differences in experimental scale, task scope, and evaluation methodology that we believe are important to highlight.
>
> ### Key distinctions
>
> We summarize the main differences in the table below:
>
> | **Dimension** | **ADMIRE-BayesOpt (Our Work)** | **MFMS-GP (Yen et al., 2025)** |
> |--------------|--------------------------------|-------------------------------|
> | **Model Scale Range** | **1M to 7B parameters**, demonstrating scalability across multiple orders of magnitude | Limited to models **under 1B parameters** |
> | **Task Coverage** | **Both pre-training AND instruction finetuning**; validated our method in each regime | Pre-training only |
> | **Evaluation Methodology** | Mixture optimization methods compared on **actual downstream task performance** from fully trained models across **diverse benchmarks**, reflecting realistic LLM development workflows | Mixture optimization methods' performance metrics **estimated and compared using trained performance predictors** rather than actual model evaluation, which may **introduce prediction bias** and not fully capture the complex relationship between data mixtures and downstream capabilities |
> | **Empirical Performance** | **Consistently achieves optimal compute-performance tradeoffs** across pre-training and instruction finetuning, demonstrating superior performance compared to baseline mixture selection strategies | Demonstrates effectiveness of the proposed multi-fidelity approach within their experimental framework |
> | **Further Insights and Analysis** | **Extensive ablations and insights**: transferability across scales (Section 6.1.3), sensitivity to model capacity (Section 5.2), failure case analysis and out-of-distribution evaluation (Section 5.1) | Primary focus on validating empirical effectiveness of the proposed multi-fidelity method |
> | **Open Research Resources** | **ADMIRE IFT Runs dataset**: 460 full training runs representing 13,000+ GPU hours made publicly available to the community | Not specified |
>
>
> We believe these contributions—particularly the **broader scope, realistic evaluation protocol, and extensive empirical insights**—represent a distinct and complementary approach that advances the practical applicability of Bayesian optimization for data mixture selection in modern LLM training pipelines. We would be happy to expand this discussion in the manuscript to make these distinctions clearer for readers.

---

### Author Response · Authors · 2025-11-08
**Revised Manuscript**

We would like to sincerely thank the reviewers for their invaluable feedback during the review process. **We have carefully incorporated all changes promised during our rebuttal in our revised manuscript**, addressing all suggestions provided.

To facilitate a swift and efficient review of these revisions, we highlight the key changes below:

- Section 6.3.3: We have added **comprehensive ablation studies** examining our design choices for both kernel functions and acquisition functions
- Section 6.3.2: We have included a detailed **computational cost breakdown**, showing that ADMIRE-BayesOpt overhead accounts for less than 0.01% of total cost.
- Section 7: We have added a **dedicated Discussion and Limitations** section that explicitly addresses future research directions, including: (a) more sophisticated kernel designs for modeling cross-scale performance relationships, (b) extensions to dynamic environments with evolving data sources, and (c)theoretical analysis and convergence guarantees for our multi-fidelity framework.
- Appendix B: We have expanded the appendix to include **detailed implementation specifics** for acquisition function optimization.
- Figures 5-7 captions: We have **improved figure captions** to provide clearer explanations of experimental results and methodology.

We believe these revisions have substantially strengthened the manuscript, and we are grateful for the reviewers' feedback, which has been instrumental in improving the quality and clarity of our work.

---

### Decision · Action_Editor_C4tM · 2025-11-25

**Recommendation:** Accept as is

**Audience:**

Yes

**Audience Explanation:**

The effective method, interesting observation, and discussion would be interesting for the ML community.

**Claims And Evidence:**

Yes

**Claims Explanation:**

The claims of the paper have been well supported by the emprical evaluation results.